# Few-shot Text Adversarial Attack for Black-box Multi-task Learning

## Abstract

Current multi-task adversarial text attacks rely on white-box access to shared internal features and assumption of homogeneous multi-task learning framework. As a result, these attacks are less effective against practical scenarios involving black-box feedback APIs and multi-model multi-task learning. To bridge this gap, we introduce **C**luster and **E**nsemble **M**util-task Text Adversarial **A**ttack (**CEMA**), an effective black-box attack that exploits the transferability of adversarial texts. Specifically, we initially employ cluster-oriented substitute model training, as a plug-and-play framework, to simplify complex multi-task scenarios into more manageable text classification attacks and train the substitute model. Next, we generate multiple adversarial candidate examples by applying various adversarial text classification methods. Finally, we select the adversarial example that attacks the most substitute models as the final attack output. CEMA is evaluated on two primary multi-task objectives: text classification and translation. In the classification task, CEMA achieves attack success rates that exceed 60% while reducing the total number of queries to 100. For the text translation task, the BLEU scores of both victim texts and adversarial examples decrease to below 0.36 with 100 queries even including the commercial translation APIs, such as Baidu Translate and Ali Translate.

## 1 Introduction

A multi-task textual adversarial attack misleads multiple tasks simultaneously through small perturbations, increasing attack efficiency and impact. It poses significant risks to safety-critical systems, leading to wrong decisions. Defending against such attacks is challenging due to the need for multi-task robustness, making it a key issue in AI security (Liu et al., 2017; Lin et al., 2022).

Research on text multi-task adversarial examples typically concentrates on tasks of the same type, particularly classification tasks (Liu et al., 2017). However, in real-world applications, multi-task learning often involves tasks of different types. Existing adversarial attack methods generally assume that attackers have access to the model architecture and shared layer information within a unified model (Guo et al., 2020). However, most commercial and application-based models are proprietary, with their architecture and parameters hidden from external attackers. Additionally, current multi-task adversarial attack strategies primarily target models that employ a shared parameter approach for managing multiple tasks. In contrast, multi-model multi-task learning approaches (Aoki et al., 2022) handle each task with a separate model, without direct parameter sharing. As a result, most existing adversarial methods, designed to attack shared parameter models, are ineffective against these systems because of the absence of a common layer to target.

Our goal is to perform multi-task textual adversarial attacks in realistic scenarios. Based on the previous analysis, such a scenario should encompass a variety of tasks, with black-box model feedback being more reflective of real-world conditions. Moreover, both parameter-sharing multi-task learning systems and multi-model multi-task learning systems must be considered. Additionally, limiting the number of queries is essential to conserve resources and reduce the risk of detection, making it a key aspect of practical attack scenarios. Therefore, this paper is driven by the following research questions(RQ):

> **RQ1**: Can attackers craft the adversarial examples with a **black-box** multi-task learning model?
> **RQ2**: How to craft the adversarial examples in **multi-model multi-task learning** model?
> **RQ3**: How to craft the adversarial examples in **few-shot** queries?
> **RQ4**: How to craft the adversarial examples in the mutil-task learning systems that **encompass a variety of tasks**?

In limited-access scenarios, a straightforward strategy is the transfer attack, which crafts adversarial examples in a substitute model. Training effective substitute models becomes challenging in the absence of a well-trained substitute model, particularly in multi-task learning, where limited access to input-output pairs and poor transferability between tasks in multi-model settings pose significant difficulties. Rather than mimicking the entire multi-task model, we propose focusing on building a substitute model with **strong discriminability**. This approach allows a single substitute model to generate adversarial examples that target all tasks simultaneously, even when trained with limited data.

We propose CEMA (**C**luster and **E**nsemble **M**ulti-task Text **A**dversarial Attack), a framework that leverages a small set of auxiliary texts sharing characteristics with the victim's texts. Using a pre-trained model, we vectorize texts and their outputs, perform clustering, and train substitute models on these auxiliary texts and cluster labels. This converts the multi-task attack into a single-task text classification problem. Repeating this process, we can obtain multiple substitute models. During the adversarial example generation phase for victim texts, for each victim text, adversarial candidates are generated for each victim text. The final adversarial example is selected based on its success across the most substitute models.

Although the substitute model trained by CEMA differs from the victim model trained through multi-task learning, our substitute model, demonstrates **strong discriminative capability**. For task $A$, if an adversarial attack on the substitute model $f^{\text{sub}}$ successfully changes the cluster label of text $x_i$ from 0 to 1, the label $y_i^A$ shifts accordingly, indicating a successful attack on task $A$. We derive and demonstrate that adversarial examples based on cluster labels, when effective against multiple substitute models, can also transfer effectively to other tasks $B, C, \ldots, N$.

During the experiment, we focus on text classification and translation within a multi-task learning framework. For the text classification task, CEMA achieves an attack success rate (ASR) of over 60% with only 100 queries. In the text translation task, CEMA reaches a BLEU score of 0.14. Even with limited auxiliary data that differs significantly from the training dataset, CEMA maintains an ASR of up to 66.40% for classification tasks and a BLEU score of 0.27 for translation tasks. The primary **contributions** are summarized as follows: ❶ To the best of our knowledge, we are the first to extend text adversarial attacks to the multi-task setting by training cluster-oriented substitute models and employing transferability-oriented adversarial example selection. The proposed CEMA method generates high-quality adversarial examples for multiple tasks simultaneously with **very few queries** in black-box and multi-model multi-task learning scenarios. ❷ We present the first *plug-and-play framework* that converts a multi-task attack into a single-task attack, enabling traditional methods to be easily adapted to multi-task scenarios. Furthermore, our approach overcomes the limitations of existing multi-task attack methods, which depend on shared layers in multi-task models. CEMA effectively handles multi-task scenarios with multi-models, whether they involve related or independent tasks. Additionally, we derive a theoretical lower bound for CEMA's success rate, showing that the probability of success increases with the number of substitute models used. ❸ We demonstrate the effectiveness of CEMA through rigorous mathematical derivations, as well as comprehensive experiments. The experimental results show the proposed CEMA achieves an attack success rate (ASR) of over 60% in text classification tasks and a BLEU score of less than 0.15 in translation tasks, indicating effective adversarial attack performance in both cases.

## 2 Preliminary

### 2.1 Transferability and Transfer attacks

**Transfer attacks** leverage adversarial examples to target different models without requiring direct access, posing a significant security threat in black-box scenarios (Szegedy et al., 2014; Papernot et al., 2017; Dong et al., 2018; Tramèr et al., 2017). **Transferability** refers to the phenomenon

where adversarial examples crafted for one model can successfully compromise other models as well (Zhang et al., 2020). Notably, several existing studies increase the amount of data available to attackers (Mahmood et al., 2021a) or generate synthetic data (Zhou et al., 2020), significantly advancing the development of transfer attacks. Meanwhile, Mahmood et al. (2021b) improve the transferability and robustness of Vision Transformers to adversarial examples

## 2.2 MULTI-TASK LEARNING AND MULTI-MODEL MULTI-TASK LEARNING

**Multi-Task Learning (MTL)** involves simultaneously training multiple related tasks, enabling models to share knowledge and improve generalization, particularly when data is limited. MTL has been extensively applied in fields such as natural language processing and computer vision, resulting in more robust models. However, challenges such as task interference and balancing shared information across tasks remain. Recent advancements seek to mitigate these challenges and enhance MTL's overall effectiveness. **Multi-Model Multi-Task Learning** extends the traditional MTL framework by utilizing separate models for each task, providing greater flexibility and better handling of task heterogeneity. This approach minimizes negative transfer and allows for task-specific optimizations. However, it also increases computational complexity and the difficulty of integrating outputs from different models. Current research focuses on hybrid methods that balance task specialization with shared learning, aiming to optimize model architectures and enhance resource efficiency.

## 3 THREAT MODEL

❶Victim Model: In this paper, we explore a more practical scenario of Multi-Model Multi-Task Learnin, focusing on the tasks of text classification and translation. We utilize publicly available APIs from the Hugging Face platform as the victim models for our attacks. Specifically, we target the SST5 and Emotion datasets for text classification, and we select DistilBERT and RoBERTa models trained on these datasets, referred to as dis-sst5, ro-sst5, dis-emotion, and ro-emotion, respectively. For the translation task, we target the opus-mt model for English-to-Chinese translation and the t5-small model for English-to-French translation. The URLs of these models are provided in Table 8 in the Appendix. Meanwhile, to simulate a more realistic attack scenario, we employ two commercial translation APIs: Baidu Translate for English-to-French translation and Ali Translate for English-to-Chinese translation. We design three multi-task victim models using these base models. **Victim Model A** comprises two classification models and one translation model: dis-sst5, dis-emo, and opus-mt. **Victim Model B** also comprises two classification models and one translation model: ro-sst5, ro-emo, and t5-small. **Victim Model C** consists of two commercial translation APIs: Baidu Translate and Ali Translate. ❷**Attacks's Goal** The goal of our attack is to degrade the performance of all tasks in a multi-task model. Adversarial examples are crafted to universally disrupt multiple tasks, not just a single one. For text classification, the objective is to ensure differing output labels between the original and adversarial inputs (*i.e.*, $y_{\mathrm{adv}} \neq y_{\mathrm{ori}}$). For translation tasks, the aim is to induce significant semantic divergence, minimizing BLEU scores between the original and adversarial outputs (*i.e.*, $\arg\min, \mathrm{BLEU}(y_{\mathrm{adv}}, y_{\mathrm{ori}})$). ❸ **Adversary Capabilities**: We analyze the adversary's capabilities from three perspectives: query access, API feedback, auxiliary data, and similarity constraint. (1) **Query Access:** Query access refers to the adversary's ability to interact with the target model before delivering the final adversarial input. We assume the attacker has up to 100 opportunities to query the victim model, with each query generating output results for all tasks. (2) **API feedback:** In a practical multi-task text adversarial attack, the attacker has no access to the internal information of the model and can only obtain the final output results of the model. Therefore, the API feedback serves as a black-box response, providing predicted labels for the classification task and the translated text (e.g., French output for English-to-French translation). (3) **Auxiliary Data:** From the perspective of data quantity, we assume that the attacker can acquire only a limited amount of Auxiliary Data, specifically 100 unlabeled texts. Regarding data distribution, we explore two scenarios: (a) The 100 unlabeled texts are sampled from the same distribution as the victim's texts, such as the 100 unlabeled texts in the validation dataset. (b) The 100 unlabeled texts and the victim's texts come from datasets of the same nature but with different distributions. (4) **Similarity Constraint:** To enhance the stealthiness of attacks, textual adversarial samples are constrained to maintain high similarity to the original text, ensuring semantic and structural coherence. This approach balances modification extent with attack effectiveness, preserving fluency and alignment with the original texts. In this

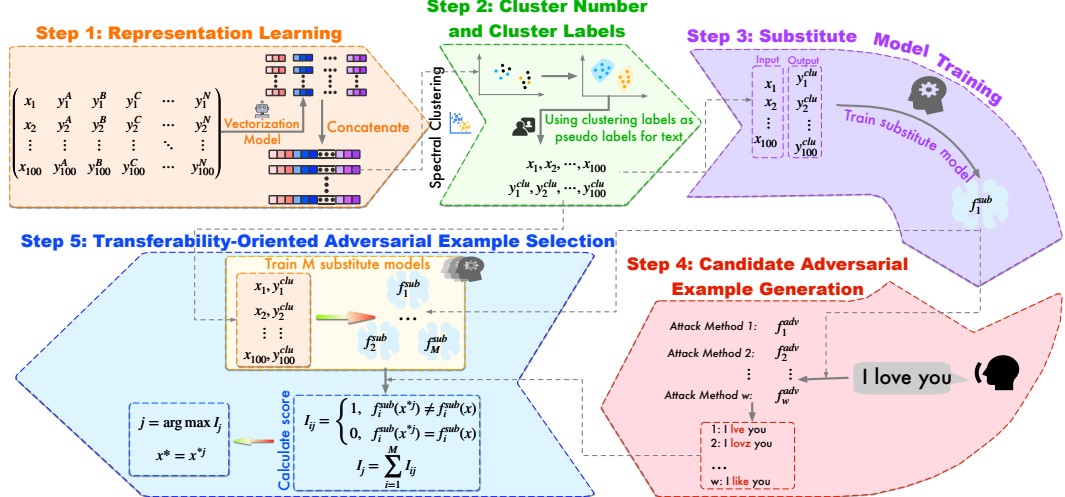

Figure 1: **The Overview of CEMA.** ❶ CEMA assigns cluster labels to auxiliary texts through a clustering method. These text-label pairs are then used to train the substitute model. This process allows CEMA to efficiently transform a multi-task scenario into a single-task text classification scenario, with only 100 queries to the black-box multi-task model. ❷ To improve attack effectiveness, CEMA applies multiple attack methods to the substitute models, generating candidate adversarial examples to refine the selection process. CEMA also trains several substitute models, selecting the final adversarial example based on its success across the majority of them.

study, we enforce a similarity threshold of 0.85, computed using the Universal Sentence Encoder (USE), a widely adopted method for reliable similarity measurement in text adversarial tasks.

## 4 METHOD

As shown in Figure 1, our method, CEMA, consists of the following steps: ❶ **Representation Learning** (Section 4.1). We convert the auxiliary texts and their outputs from multiple tasks into vector form using representation learning. ❷ **Clustering to Generate cluster labels** (Section 4.2). After determining the optimal number of clusters, we apply a clustering algorithm to the vector representations of the auxiliary texts and their outputs, assigning a cluster label to each auxiliary text. ❸ **Training Substitute Models** (Section 4.3). We train substitute models $f^{\text{sub}}$ using auxiliary texts as input and their corresponding cluster labels as output. ❹ **Generation of Adversarial Candidates** (Section 4.4). We apply various text adversarial attack methods to the substitute model $f^{\text{sub}}$ to generate multiple adversarial candidates. ❺ **Final Adversarial Example Selection** (Section 4.5). By repeating steps ❶, ❷, and ❸, we can train multiple substitute models, $f_1^{\text{sub}}, f_2^{\text{sub}}, \ldots, f_M^{\text{sub}}$. We select the adversarial candidate that successfully attacks the most substitute models as the final adversarial example.

### 4.1 REPRESENTATION LEARNING

In a multi-task model, both the input text and output labels need to be appropriately vectorized to effectively capture the relevant information. This section details the specific approach used for learning the representations of input text and output labels. Pre-trained models are extensively utilized in NLP for textual feature extraction (Tabassum & Patil, 2020; Han et al., 2021). These models are highly effective as they are trained on large-scale datasets, enabling them to learn general language patterns and representations. Furthermore, concatenating multiple text representations allows for the simultaneous encoding of multiple texts (Devlin et al., 2019). Accordingly, we leverage a pre-trained model to vectorize both the input text and output labels, generating their respective embeddings. These embeddings are subsequently concatenated to form a unified representation that captures the information from both the input text and the output labels.

---

**Algorithm 1:** The substitute model Training Process

---

**Input:** The dataset to be attacked $\boldsymbol{D} = \{x_1, x_2, \cdots, x_n\}$, where $x_i$ is the input text; embedding function $f_E$; clustering function $f_c$; number of clusters $k$; training epoch $e_{\max}$; targeted model $f_t$

**Output:** The substitute model $f^{\text{sub}}$

1 **for** $i = 1$ to $n$ **do**
2     $y_i^A, y_i^B, \ldots, y_i^N = f_t(x_i)$     ▶ Input $x_i$ to the targeted model $f_t$ to obtain the corresponding label $\boldsymbol{y_i^t}$
3     $\boldsymbol{E}(x_i) = f_E(x_i)$ ; $\boldsymbol{E}_{y_i^J} = f_{\text{pre}}(y_i^J)$
4     $\boldsymbol{E}_i = \text{Concat}(\boldsymbol{E}_{x_i}, \boldsymbol{E}_{y_i^A}, \ldots, \boldsymbol{E}_{y_i^N})$
5 $\boldsymbol{E}_{\text{all}} = [\boldsymbol{E}_1, \boldsymbol{E}_2, \cdots, \boldsymbol{E}_n]$     ▶ Representation learning
6 Perform a cluster analysis on $\boldsymbol{E}_{\text{all}}$ and refine the internal parameters of the clustering model $f_c$
7 **for** $i = 1$ to $n$ **do**
8     Input $\boldsymbol{E}_i$ into the clustering algorithm to generate the corresponding pseudolabel $y_i^{\text{pse}} = f_c(\boldsymbol{E}_i)$     ▶ Obtaining cluster labels and pseudo labels
9 The victim text cluster label pairs data: $\boldsymbol{PD} = \{(x_1, y_1^{\text{pse}}), (x_2, y_2^{\text{pse}}), \cdots, (x_n, y_n^{\text{pse}})\}$
10 **for** $i = 1$ to $e_{\max}$ **do**
11     Train the substitute model $f^{\text{sub}}$ on $\boldsymbol{PD}$ to adjust the parameters $\theta_{f^{\text{sub}}}$: $\theta_{f^{\text{sub}}} \leftarrow \text{train}(f^{\text{sub}}, \boldsymbol{PD})$     ▶ Train substitute model
12 **return** The substitute model $f^{\text{sub}} = f^{\text{sub}}(\boldsymbol{PD}; \theta_{f^{\text{sub}}})$

---

As outlined in lines 1-5 of Algorithm 1, we begin by querying the multi-task model to retrieve the output text for each task. Next, auxiliary text with corresponding output results are vectorized by pre-trained models to extract relevant features for subsequent clustering process. We define the multi-task model as $f_v$, which deals with the set of tasks $A, B, \ldots, N$. The pre-trained model is defined as $f_{\text{pre}}$. The attacker is assumed to have access to a small set of auxiliary texts $\boldsymbol{X}$, which share the same distribution as the victim texts. For each auxiliary text $x_i$ in $\boldsymbol{X}$, we query $f_v$ to obtain the corresponding outputs $y_i^A, y_i^B, \ldots, y_i^N$. Next, we use the pre-trained model $f_{\text{pre}}$ to vectorize $x_i$ and $\{y_i^A, y_i^B, \ldots, y_i^N\}$, resulting in the vectors $\{\boldsymbol{E}_{x_i}, \boldsymbol{E}_{y_i^A}, \ldots, \boldsymbol{E}_{y_i^N}\}$. These vectors are concatenated to form the final vector $\boldsymbol{E}_i$, representing $x_i$ and its outputs $y_i^A, y_i^B, \ldots, y_i^N$. Thus, $\boldsymbol{E}_i$ is defined as follows:

$$\boldsymbol{E}_{x_i} = f_{\text{pre}}(x_i), \boldsymbol{E}_{y_i^J} = f_{\text{pre}}(y_i^J), \boldsymbol{E}_i = Concat(\boldsymbol{E}_{x_i}, \boldsymbol{E}_{y_i^A}, \ldots, \boldsymbol{E}_{y_i^N}), \tag{1}$$

where the $Concat$ indicates the concatenation of $\{\boldsymbol{E}_{x_i}, \boldsymbol{E}_{y_i^A}, \ldots, \boldsymbol{E}_{y_i^N}\}$.

## 4.2   CLUSTER NUMBER AND CLUSTER LABELS

In Section 4.1, we obtain the representations for each text input and output. We then perform a clustering analysis on these representations, with the number of clusters being a crucial parameter. Before clustering, we determine the optimal number of clusters by selecting the value that maximizes strong discriminative capability for each cluster group. When the number of clusters is 2, the two clusters can be interpreted as class $C_A$ and $\overline{C_A}$. (Boongoen & Iam-On, 2018). Therefore, we set the number of clusters to 2. After determining the number of clusters to be 2, we perform clustering analysis on the 100 vectors using the Spectral clustering method (Zhang et al., 1996). For each vector $\boldsymbol{E}_i$, we derive its corresponding cluster label $y_i^{\text{clu}}$, which is later assigned as the pseudolabel for $x_i$. With only 100 unlabeled texts, we cannot fully capture the dataset's distribution. Therefore, we perform multiple clustering runs and select the result that best approximates a uniform distribution by maximizing entropy (i.e., ensuring the cluster sizes are as close to 50 as possible). The proof of entropy maximization for a uniform distribution is in Section B of the appendix.

## 4.3   SUBSTITUTE MODEL TRAINING

Once the cluster labels are obtained, we employ the auxiliary texts paired with their respective cluster labels to train a substitute model. This approach effectively converts the multi-task text adversarial

attack scenario into a conventional text classification adversarial attack scenario. The substitute model $f^{\text{sub}}$ is trained with the auxiliary texts serving as input data and the cluster labels as the corresponding output labels. The process is shown in lines 10-12 of Algorithm 1. More details about the substitute model architecture and substitute model training are presented in Appendix D.

### 4.4 CANDIDATE ADVERSARIAL EXAMPLE GENERATION

Once the substitute model is generated, we apply several adversarial text attack methods to $f^{\text{sub}}$. These methods produce multiple adversarial examples. We then define criteria to select the final adversarial examples from the candidates generated. In this section, we begin by explaining the importance of generating multiple adversarial candidate examples. We assume that $m$ adversarial text attack methods are used to generate $m$ adversarial examples on the substitute model $f_1^{\text{sub}}$. These adversarial examples are denoted as $x_i^{*1}, x_i^{*2}, \ldots, x_i^{*m}$. Each example has a corresponding probability of successfully attacking the victim model, denoted as $p_i^{*1}, p_i^{*2}, \ldots, p_i^{*m}$. The minimum probability among these is denoted as $p_{\min}^*$, where $p_{\min}^* = \min(p_i^{*1}, p_i^{*2}, \ldots, p_i^{*m})$. We calculate the probability, $p_s$, that at least one of these adversarial examples successfully attacks the victim model as follows:

$$p_s = 1 - (1 - p_i^{*1})(1 - p_i^{*2}) \cdots (1 - p_i^{*m}) = 1 - \prod_{j=1}^{m}(1 - p_i^{*j}) \tag{2}$$

We analyze the trend of $p_s$ as the number of adversarial examples, $m$, increases.

$$\begin{aligned} p_s &= 1 - (1 - p_i^{*1})(1 - p_i^{*2}) \cdots (1 - p_i^{*m}) \\ &\geq 1 - (1 - p_{\min}^*)(1 - p_{\min}^*) \cdots (1 - p_{\min}^*) = 1 - (1 - p_{\min}^*)^m \end{aligned} \tag{3}$$

As $m$ increases, the probability $(1 - p_{\min}^*)^m$ decreases and approaches 0. Conversely, the probability $1 - (1 - p_{\min}^*)^m$ increases and approaches 1. Since $p_s$ is a probability, it must satisfy $0 \leq p_s \leq 1$ (Kolmogoroff, 1933). Combining this result with equation (3), we derive the following formula:

$$1 - (1 - p_{\min}^*)^m \leq p_s \leq 1 \tag{4}$$

As $m$ increases towards infinity, equation (4) undergoes the following changes:

$$\lim_{m \to \infty} 1 - (1 - p_{\min}^*)^m = 1, \text{then } 1 \leq p_s \leq 1, \text{which means } p_s = 1. \tag{5}$$

Equation (5) demonstrates that as $m$ approaches infinity, the probability of a successful attack reaches 100%. In contrast, (3) illustrates that the attack success rate increases gradually with the growth of $m$. These findings emphasize the necessity and importance of generating multiple adversarial candidate examples.

**Remark** *The previous analysis assumes independence. In Section T of the appendix, we examine the case of non-independence. We find that, in the non-independent scenario, using more methods to generate adversarial examples increases the likelihood of successfully attacking the victim model.*

### 4.5 TRANSFERABILITY-ORIENTED ADVERSARIAL EXAMPLE SELECTION

In Section 4.4, we demonstrate that generating additional adversarial candidate examples increases the likelihood of finding a successful adversarial example, which can then effectively attack the victim model. This section focuses on the process of selecting the most likely successful adversarial example from the generated candidates. We explore the criteria and methods used to identify the most effective example.

We first select the adversarial candidate with the highest transferability as the final example. To evaluate transferability, we train multiple substitute models and count the number of successful attacks against them. Ultimately, we choose the adversarial candidate that successfully attacks the most substitute models as the final adversarial example. The detailed steps are presented as follows: ❶ **Training Multiple Substitute Models:** We randomly sample 80% of the 100 auxiliary text-cluster label pairs to form the training set for a new substitute model. This process is repeated $w$ times, yielding $w$ substitute models, denoted as $f_1^{\text{sub}}, f_2^{\text{sub}}, \ldots, f_w^{\text{sub}}$. ❷ **Calculating the Transferability Score:** For each victim text $x_k$, we generate $m$ adversarial candidate examples, denoted as $\{x_k^{*1}, x_k^{*2}, \ldots, x_k^{*m}\}$. The transferability score for $x_k^{*j}$ is calculated as follows:

$$I_{kij} = \begin{cases} 1, & f_i^{\text{sub}}\left(x_k^{*j}\right) \neq f_i^{\text{sub}}(x_k); \\ 0, & f_i^{\text{sub}}\left(x_k^{*j}\right) = f_i^{\text{sub}}(x_k); \end{cases} \quad I_{kj} = \sum_{i=1}^{w} I_{kij} \quad j = \arg\max_j I_{kj}. \quad (6)$$

where $f_i^{\text{sub}}\left(x_k^{*j}\right)$ represents the output label of $x_k^{*j}$ is produced by the substitute model $f_i^{\text{sub}}$. Similarly, $f_i^{\text{sub}}(x_k)$ is the output label of $x_k$ generated by the same model. If $f_i^{\text{sub}}\left(x_k^{*j}\right) \neq f_i^{\text{sub}}(x_k)$, then $x_k^{*j}$ successfully attacks the substitute model $f_i^{\text{sub}}$. Therefore, $I_{kj}$ measures the number of substitute models that $x_k^{*j}$ successfully attacks. The adversarial example that successfully attacks the largest number of substitute models is then selected as the final adversarial example. In other words, adversarial examples capable of attacking multiple substitute models demonstrate greater transferability and higher probability of successfully attacking the victim model $f_v$.

## 5 EXPERIMENT

### 5.1 EXPERIMENT SETUP

**Dataset:** We evaluate the effectiveness of our method using the **SST5** and **Emotion** datasets. The **Emotion** dataset, containing six emotions, is sourced from Twitter. The **SST5** dataset, used for sentiment analysis, includes five categories from movie reviews. Detailed statistics are provided in Appendix F, Table 7. **Baselines:** Since no prior black-box text adversarial attack focuses on multi-task scenarios, we select traditional textual attack methods. For text classification, we use BAE (Garg & Ramakrishnan, 2020), FD (Papernot et al., 2016), Hotflip (Ebrahimi et al., 2018b), SememePSO (Zang et al., 2020), and TextBugger (Ren et al., 2019). For text translation, we select Hotflip (Trans) (Ebrahimi et al., 2018b), kNN (Michel et al., 2019), Morphin (Tan et al., 2020), RA (Zou et al., 2019), Seq2Sick (Cheng et al., 2020), and TransFool (Sadrizadeh et al., 2023). CEMA operates with substantially fewer queries. For a fair comparison, we limit all baseline methods to 30 final queries when attacking the target text. Preliminary details about these methods are listed in Tables 9a and 9b in Appendix H. **Metrics:** We use the following metrics to evaluate our method: ❶ **ASR (Attack Success Rate):** A higher ASR indicates a more effective attack. ❷ **Average Query:** Fewer queries suggest a better attack method. ❸ **BLEU (Bilingual Evaluation Understudy):** A lower BLEU score signifies a more successful disruption of translation quality.

### 5.2 COMPARISON OF RESULTS BETWEEN CEMA AND BASELINES

Given the absence of multi-task adversarial methods for black-box outputs in translation tasks, we compare the CEMA method with existing adversarial techniques for text translation and classification. The results, presented in Table 1 and Table 2, demonstrate that CEMA achieves state-of-the-art (SOTA) performance in the SST5 and Emotion datasets across the victim models $A$, $B$, and $C$. **For each dataset,** 100 **queries are made per task, with SST5 containing** $2,210$ **texts and Emotion** $2,000$**, averaging** $0.045$ **and** $0.05$ **queries per task, respectively.** Remarkably, in this black-box, low-access scenario, CEMA achieved an ASR of over $59\%$ on classification tasks, with a maximum of $80.80\%$. Furthermore, in translation tasks, CEMA's BLEU score was below $0.16$, outperforming the second-best method by a considerable margin. CEMA also achieved SOTA results against the victim model $C$ (Baidu and Ali Translate) using only 100 auxiliary texts. As commercial translators are closed-source, we compared the black-box attack algorithms Morphin and TransFool. CEMA consistently outperformed the second-best attack algorithm, with BLEU scores below $0.35$, using just 100 queries.

### 5.3 THE IMPACT OF CLUSTER NUMBER

In CEMA, we use two clusters. To assess the impact of increasing the number of clusters, we also conducted experiments with three and four clusters. As illustrated in Figure 2, increasing the number of clusters reduces attack performance. When the number of clusters increased from 2 to 4, the average ASR decreased from 58.83% and 64.55% to 46.20% and 52.10%, respectively, while the average BLEU score increased from 0.16 and 0.18 to 0.41 and 0.32. Clearly, the best attack

Table 1: The attack performance of CEMA. Text classification tasks use the ASR(%)↑ metric, while text translation tasks use the BLEU↓ metric. Other adversarial attack methods can only be applied to their specific tasks, whereas CEMA simultaneously attacks all tasks.

| Dataset | SST5 | | | | Emotion | | | |
|---|---|---|---|---|---|---|---|---|
| Victim Model | Victim Model A | | Victim Model B | | Victim Model A | | Victim Model B | |
| Text Classification | dis-sst5 (A) | | ro-sst5 (B) | | dis-sst5 (A) | | ro-sst5 (B) | |
| Metric | ASR(%)↑ | Queries↓ | ASR(%)↑ | Queries↓ | ASR(%)↑ | Queries↓ | ASR(%)↑ | Queries↓ |
| Bae | 42.71 | 21.43 | 39.14 | 21.48 | 31.55 | 26.98 | 28.50 | 25.31 |
| FD | 25.20 | 12.56 | 22.30 | 9.71 | 47.10 | 29.88 | 20.75 | 12.09 |
| Hotflip | 41.50 | 11.52 | 29.03 | 11.74 | 46.85 | 9.80 | 41.65 | 10.14 |
| PSO | 45.14 | 11.04 | 41.50 | 12.38 | 46.05 | 8.92 | 44.95 | 8.94 |
| TextBugger | 30.36 | 31.46 | 20.85 | 30.32 | 35.10 | 11.41 | 29.40 | 11.37 |
| Leap | 32.55 | 9.75 | 30.07 | 9.54 | 26.30 | 7.01 | 15.50 | 6.93 |
| CT-GAT | 29.37 | 20.92 | 24.80 | 37.54 | 25.90 | 21.42 | 26.75 | 21.33 |
| HQA | 46.11 | 29.35 | 39.64 | 29.08 | 37.35 | 29.74 | 35.85 | 21.47 |
| **CEMA** | **73.57** | **0.045** | **75.66** | **0.045** | **80.80** | **0.05** | **60.40** | **0.05** |
| Text Classification | dis-emotion (A) | | ro-emotion (B) | | dis-emotion (A) | | ro-emotion (B) | |
| Metric | ASR(%)↑ | Queries↓ | ASR(%)↑ | Queries↓ | ASR(%)↑ | Queries↓ | ASR(%)↑ | Queries↓ |
| Bae | 39.81 | 27.33 | 14.65 | 28.06 | 32.25 | 21.84 | 32.95 | 21.83 |
| FD | 35.43 | 29.22 | 9.55 | 16.54 | 22.30 | 12.81 | 17.50 | 18.43 |
| Hotflip | 33.39 | 10.86 | 22.80 | 12.28 | 29.00 | 14.28 | 28.05 | 14.40 |
| PSO | 41.90 | 9.02 | 35.25 | 9.45 | 39.50 | 11.83 | 37.65 | 12.10 |
| TextBugger | 30.00 | 11.35 | 40.95 | 11.35 | 20.85 | 30.32 | 21.45 | 30.33 |
| Leap | 21.00 | 6.93 | 26.00 | 7.01 | 40.58 | 9.73 | 37.65 | 9.78 |
| CT-GAT | 39.32 | 21.36 | 33.45 | 21.49 | 28.10 | 26.06 | 30.85 | 25.34 |
| HQA | 37.76 | 21.44 | 31.95 | 29.44 | 37.40 | 22.44 | 36.40 | 23.16 |
| **CEMA** | **62.27** | **0.045** | **64.01** | **0.045** | **65.40** | **0.05** | **59.6** | **0.05** |
| Text Translation | opus-mt(en-zh) (A) | | t5-small(en-fr) (B) | | opus-mt(en-zh) (A) | | t5-small(en-fr) (B) | |
| Metric | BLEU↓ | Queries↓ | BLEU↓ | Queries↓ | BLEU↓ | Queries↓ | BLEU↓ | Queries↓ |
| Hot-trans | 0.24 | 9.76 | 0.24 | 9.45 | 0.20 | 9.36 | 0.19 | 9.81 |
| KNN | 0.31 | 6.19 | 0.31 | 6.19 | 0.61 | 13.34 | 0.28 | 6.08 |
| Morphin | 0.30 | 6.79 | 0.37 | 11.1 | 0.27 | 5.06 | 0.22 | 3.84 |
| RA | 0.25 | 3.18 | 0.19 | 4.26 | 0.23 | 2.79 | 0.21 | 2.11 |
| Seq2sick | 0.38 | 4.45 | 0.46 | 6.05 | 0.62 | 7.09 | 0.29 | 4.05 |
| TransFool | 0.77 | 3.32 | 0.44 | 3.91 | 0.81 | 3.89 | 0.67 | 3.58 |
| **CEMA** | **0.14** | **0.045** | **0.18** | **0.045** | **0.15** | **0.05** | **0.23** | **0.05** |

Table 2: Attack performance of different methods on victim model C. Victim model C consists of two commercial closed-source translation models, namely Alibaba Translate and Baidu Translate.

| Data | Victim Model C | Baidu Translate (en-fr) (C) | | Ali Translate (en-zh) (C) | |
|---|---|---|---|---|---|
| | Methods | BLEU↓ | Queries↓ | BLEU↓ | Queries↓ |
| SST5 | Morphin | 0.54 | 40.48 | 0.60 | 48.45 |
| | TransFool | 0.51 | 23.53 | 0.59 | 31.20 |
| | **CEMA** | **0.29** | **0.045** | **0.15** | **0.045** |
| Emotion | Morphin | 0.40 | 27.79 | 0.55 | 12.70 |
| | TransFool | 0.36 | 12.70 | 0.49 | 30.91 |
| | **CEMA** | **0.35** | **0.05** | **0.29** | **0.05** |

performance is achieved when using two clusters. As discussed in Section 4.2, *two clusters provide the highest discriminative ability and optimal attack performance in the binary-class substitute model.*

## 5.4 THE IMPACT OF CANDIDATE ADVERSARIAL EXAMPLE NUMBER

CEMA utilizes three attack methods: DWB, FD, and Textbugger. Each method generates three adversarial examples for each victim text. To assess the impact of reducing the number of examples, we conducted experiments using only Textbugger. As shown in Table 3, attack performance declines as the number of adversarial examples decreases. This reduction occurs because a smaller adversarial space leads to lower ASR and higher BLEU scores, consistent with the analysis in Appendix D. When the number of attack algorithms increases from one to three, the average ASR rises by 30.39%,

Table 3: Performance of CEMA under different number setting of candidate adversarial examples.

| Data | Example Number | Victim Model A | | | Victim Model B | | |
|---|---|---|---|---|---|---|---|
| | | dis-sst5 (A) | dis-emoton (A) | opumt(en-zh) (A) | ro-sst5 (B) | ro-emotion (B) | t5-small(en-fr) (B) |
| | | ASR(%)↑ | ASR(%)↑ | BLEU↓ | ASR(%)↑ | ASR(%)↑ | BLEU↓ |
| SST5 | 3 | 73.57 | 62.27 | 0.14 | 75.66 | 64.01 | 0.18 |
| | 1 | 50.42 | 29.23 | 0.30 | 43.79 | 24.73 | 0.35 |
| Emotion | 3 | 80.80 | 65.40 | 0.15 | 60.40 | 59.60 | 0.23 |
| | 1 | 29.20 | 34.80 | 0.31 | 39.20 | 47.20 | 0.39 |

Table 4: Performance of CEMA under various clustering methods.

| Data | Clustering Method | Victim Model A | | | Victim Model B | | | victiom Model C | |
|---|---|---|---|---|---|---|---|---|---|
| | | dis-sst5 | dis-emotion | opus-mt (en-zh) | ro-sst5 | ro-emotion | t5-small (en-fr) | Baidu Translate (en-fr) | Ali Translate (en-zh) |
| | | ASR(%)↑ | ASR(%)↑ | BLEU↓ | ASR(%)↑ | ASR(%)↑ | BLEU↓ | BLEU↓ | BLEU↓ |
| SST5 | Spectral | 73.57 | 62.27 | 0.14 | 75.66 | 64.01 | 0.18 | 0.29 | 0.13 |
| | Kmeans | 72.97 | 61.17 | 0.12 | 74.96 | 63.63 | 0.17 | 0.32 | 0.11 |
| | BIRCH | 74.27 | 62.77 | 0.09 | 73.26 | 60.57 | 0.15 | 0.23 | 0.16 |
| Emotion | Spectral | 80.80 | 65.40 | 0.15 | 60.40 | 59.60 | 0.23 | 0.35 | 0.21 |
| | Kmeans | 77.20 | 50.80 | 0.18 | 59.30 | 61.65 | 0.23 | 0.37 | 0.15 |
| | BIRCH | 76.35 | 52.65 | 0.13 | 64.01 | 56.55 | 0.27 | 0.43 | 0.21 |

while the average BLEU score decreases by 0.16. These results suggest that increasing the number of attack algorithms enhances overall attack performance.

## 5.5 THE IMPACT OF CLUSTERING METHODS

In CEMA, we use spectral clustering as the primary method. To assess the impact of different clustering techniques on experimental results, we also implement K-means (Krishna & Murty, 1999) and BIRCH clustering (Zhang et al., 1996). As shown in Figure 3 and Table 4, the ASR in the classification task shows minimal variation across clustering methods. In contrast, the BLEU score in the translation task fluctuates more significantly, but no consistent pattern emerges. No clustering method consistently achieves SOTA performance across all scenarios. The average ASR for Spectral, KMeans, and BIRCH are 67.71%, 65.21%, and 65.05%, respectively, with average BLEU scores of 0.21, 0.20, and 0.21. Therefore, we conclude that *while clustering methods do influence attack performance, their impact is largely random and does not consistently favor one method over another.*

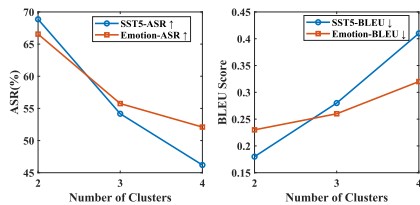

Figure 2: The average ASR and BLUE of different numbers of clusters. Fewer clusters result in better attack performance.

## 5.6 THE IMPACT OF VECTORIZATION METHODS

Given that our multi-task framework includes a translation task, we use the multilingual mT5 (Xue, 2020) for text vectorization, along with the XLM-R (Conneau, 2019) model and one-hot encoding (Rodríguez et al., 2018). One-hot encoding converts categorical data into binary vectors, where each category is represented by a unique vector with a single 1 and all other elements set to 0. To mitigate data leakage, we limit one-hot encoding to 100 samples from the additional dataset. As shown in Figure 3 and Table 5, different vectorization methods have no significant impact on attack performance in the classification task. In the translation task, while vectorization methods cause fluctuations in attack results, these variations are irregular, and no single method consistently achieves SOTA performance across all datasets and victim models. Specifically, the average ASR for the mT5, XLM-R, and one-hot vectorization methods is 67.71%, 65.81%, and 67.72%, respectively, while the average BLEU scores are 0.21, 0.22, and 0.22, respectively. Therefore, we conclude that *vectorization methods do not substantially influence attack performance.*

Table 5: Performance of CEMA under various vectorization methods.

| Data | Vectorization Method | Victim Model A | | | Victim Model B | | | Victim Model C | |
| | | dis-sst5 | dis-emotion | opus-mt (en-zh) | ro-sst5 | ro-emotion | t5-small (en-fr) | Baidu Translate (en-fr) | Ali Translate (en-zh) |
| | | ASR(%)↑ | ASR(%)↑ | BLEU↓ | ASR(%)↑ | ASR(%)↑ | BLEU↓ | BLEU↓ | BLEU↓ |
| SST5 | mT5 | 73.57 | 62.27 | 0.14 | 75.66 | 64.01 | 0.18 | 0.29 | 0.13 |
| | XLM-R | 73.55 | 61.09 | 0.17 | 74.90 | 63.44 | 0.19 | 0.38 | 0.11 |
| | one-hot | 73.57 | 61.24 | 0.11 | 75.09 | 62.90 | 0.13 | 0.23 | 0.15 |
| Emotion | mT5 | 80.80 | 65.40 | 0.15 | 60.40 | 59.60 | 0.23 | 0.35 | 0.21 |
| | XLM-R | 81.05 | 64.95 | 0.19 | 53.80 | 53.75 | 0.19 | 0.37 | 0.16 |
| | one-hot | 81.05 | 65.65 | 0.18 | 62.35 | 59.90 | 0.27 | 0.43 | 0.25 |

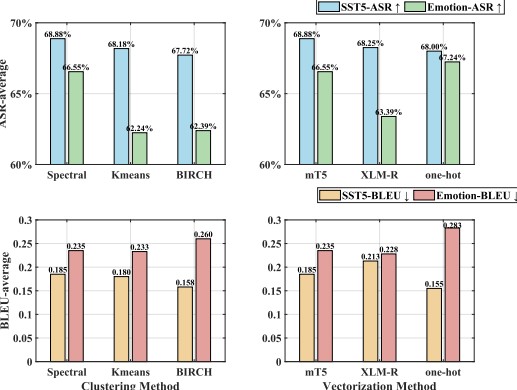

Figure 3: The average ASR and BLUE of CEMA under various clustering and vectorization methods.

Table 6: Zero-shot attack performance of CEMA.

| | | Victim Model A | | | Victim Model B | | | Victim Model C | |
| | | dis-sst5 | dis-emotion | opus-mt (en-zh) (A) | ro-sst5 (A) | ro-emotion (A) | t5-small (en-fr) | Baidu Translate (en-fr) | Ali Translate (en-zh) |
| Victim Data | Access Data | ASR(%)↑ | ASR(%)↑ | BLEU↓ | ASR(%)↑ | ASR(%)↑ | BLEU↓ | BLEU↓ | BLEU↓ |
| SST5 | SST5 | **73.57** | **62.27** | 0.14 | **75.66** | **64.01** | 0.18 | **0.29** | **0.15** |
| | Emotion | 64.00 | 60.80 | **0.18** | 59.20 | 52.00 | 0.22 | 0.36 | 0.27 |
| Emotion | Emotion | **80.80** | **65.40** | **0.15** | **60.40** | **59.60** | **0.23** | **0.35** | **0.29** |
| | SST5 | 66.40 | 36.00 | 0.21 | 48.80 | 46.40 | 0.36 | 0.44 | 0.42 |

## 5.7 ZERO-SHOT ATTACK OF CEMA

In this section, we evaluate CEMA's effectiveness under more stringent conditions, where the attacker can only access data related to the training set. Both the SST5 and Emotion datasets are related to sentiment analysis but differ significantly in label space and distribution. To test this, we used 100 unlabeled texts from the Emotion validation set as auxiliary data for the SST5 attack, and vice versa. The results in Table 6 show that, despite limited auxiliary data and significant distribution differences, CEMA achieves a 66.40% attack success rate and a BLEU score of 0.27. This suggests that an attacker needs only partial knowledge of the training data and can collect relevant data from the Internet to execute a successful attack on the multi-task system using CEMA.

## 6 CONCLUSION

In this paper, we present a more practical multi-task learning scenario where attackers can only access final black-box outputs through limited queries. To address this challenge, we propose the CEMA method, which achieves state-of-the-art (SOTA) performance in experimental evaluations with just 100 queries and black-box outputs. Furthermore, CEMA can incorporate any text classification attack algorithm, and its performance improves as the number of attack algorithms increases. In the future, we aim to extend CEMA to multi-task models across other modalities.

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

This appendix includes our supplementary materials as follows:

- Related Work in Section A.

- Derivation of the maximum entropy distribution in Section B

- More Details of Theorem and Proof in Section C

- More Details of substitute model architecture in Section D

- More Details of Substitute Model Training in Section E

- More Details of Data in Section F

- Url of victim model used in Section G

- Details of Baselines in Section H

- Performance Evaluation on Six Downstream Tasks in Section I

- Performance Evaluation on Summary and Text to Image Tasks in Section J

- Evaluation on Few-Shot Learning and Additional Model Queries in Section K

- Revisit the Transfer Attack in Section L

- Evaluation with Sim and Total-Query Metrics in Section M

- Experiment Results with More Text Classification Baselines in Section N

- Performance with Transfer Attacks in Section O

- More Details of Defense Method in Section P

- Experiment Result of Random Shuffle in Section Q

- Definition of Text Classification Adversarial Examples and NMT Adversarial Examples in Section R

- Experiment Result for Verifying Independence in Section S

- Supplementary Explanation for the Non-Independent Case in Section Candidate Adversarial Example Generation in Section T

- More Details of MMMTL in Section U

## A  RELATED WORK

### A.1  TEXT CLASSIFICATION ADVERSARIAL ATTACK

In historical textual adversarial research, the predominant methods revolve around scenarios with singular output results (Waghela et al., 2024; Han et al., 2024; Zhu et al., 2024; Kang et al., 2024). These studies focus on the techniques for morphing the original text into adversarial counterparts, including the manipulation of pivotal chars (Ebrahimi et al., 2018b; Gil et al., 2019; Ebrahimi et al., 2018a; Gao et al., 2018; Ren et al., 2019; Jin et al., 2020; Li et al., 2019), words (Wang et al., 2022; Guo et al., 2021; Meng & Wattenhofer, 2020; Sato et al., 2018; Cheng et al., 2019; Lee et al., 2022; Li et al., 2020a; Hu et al., 2024; Liu et al., 2024; 2023; Li et al., 2019) and sentence. These methods are segmented into three distinct categories based on the response from the target model, encompassing white-box attacks, soft-label black-box attacks, and hard-label black-box attacks. In white-box attacks, adversaries gain full access to all relevant information about the target model. The Hotflip (Ebrahimi et al., 2018b) sequentially replaces crucial words based on their calculated importance scores. The FD method (Papernot et al., 2016) constructs adversarial examples depending on the model's gradient information. In soft-label black-box attacks, numerous methods are geared towards disturbing the words in accordance with output probabilities (Lee et al., 2022; Maheshwary et al., 2021b; Wang et al., 2021a; Li et al., 2020a). Bert-ATTACK (Li et al., 2020a) focuses on word attacks using a refined Bert model. SememePSO (Zang et al., 2020) enhances the search landscape to construct adversarial examples. Bae (Garg & Ramakrishnan, 2020) is an attack strategy centered on BERT to replace words. Simultaneously, the DeepWordBug (DWB) method (Gao et al., 2018)

prioritizes the words for assault based on the output probabilities. Hard-label adversarial attacks present a more realistic scenario. HLGA (Maheshwary et al., 2021a) employs stochastic starting words and employs a genetic algorithm to craft adversarial examples. HQA-attack (Liu et al., 2024) starts by maximally restoring original words, reducing disruption. It then uses synonyms of remaining altered words to enhance the adversarial example.

## A.2 Neural Machine Translation Adversarial Attack

Neural Machine Translation (NMT) models, which automatically convert input sentences into translated output, have achieved remarkable results by employing deep neural networks like Transformers (Bahdanau, 2014; Vaswani, 2017). These models are now extensively used across various applications due to their high performance. However, erroneous outputs generated by NMT models can lead to significant risks, particularly in security-sensitive contexts. Recent research has explored adversarial attacks targeting NMT models to address these concerns. Character-level NMT models are highly vulnerable to character manipulations such as typos in a block-box setting (Belinkov & Bisk, 2017; Ebrahimi et al., 2018a). as well as pushing/removing words from the translation. However, character manipulations and typos are easily detected by humans or review strategies. Hence, most adversarial attacks against NLP and NMT systems use a word replacement strategy instead. Seq2sick (Cheng et al., 2020) proposes a projected gradient method combined with group lasso and gradient regularization, conducting non-overlapping attacks and targeted keyword attacks. Similarly, Transfool (Sadrizadeh et al., 2023) also uses the gradient projection method, defining a new optimization problem and linguistic constraints to compute semantic-preserving and fluent attacks against NMT models. Morphin (Tan et al., 2020) generates plausible and semantically similar adversaries by perturbing the inflections in clean examples to investigate the robustness of NLP models to inflectional perturbation. kNN (Michel et al., 2019)is a white-box untargeted attack against NMT models that substitutes some words with their neighbors in the embedding space. RG (Zou et al., 2019)investigates the issue by generating adversarial examples through a new paradigm based on reinforcement learning, which generates more reasonable tokens and secures semantic constraints.

## A.3 Mutil-task Adversarial Attack

A Multi-task Adversarial Attack is an adversarial machine learning strategy designed to generate examples that deceive multiple models or systems simultaneously (Guo et al., 2020; Ghamizi et al., 2022), rather than just one. As far as we know, there is currently no related work on multi-task adversarial attacks in the field of text. In other fields, MTA (Guo et al., 2020) is designed to generate adversarial perturbations for all three pre-trained classifiers simultaneously by leveraging shared knowledge among tasks. There is an attack method (Sobh et al., 2021) that targets visual perception in autonomous driving, which is applied in a wide variety of multi-task visual perception deep networks in distance estimation, semantic segmentation, motion detection, and object detection. MTADV (Wang et al., 2024) is a multitask adversarial attack against facial authentication, which is effective against various facial data sets.

## A.4 Transfer Attack

**Transfer attacks** leverage adversarial examples to target different models without requiring direct access, posing a significant security threat in black-box scenarios (Papernot et al., 2017; Dong et al., 2018). Then, in the absence of a substitute model, several studies demonstrate that auxiliary data can also facilitate successful attacks through training a substitute model and leveraging transfer attacks (Li et al., 2020c; Sun et al., 2022). Additionally, more effective loss functions have been proposed to train substitute models (Wang et al., 2021b; Li et al., 2020b; Naseer et al., 2019; Richards et al., 2021; Huan et al., 2020), as well as techniques to refine substitute models (Xiaosen et al., 2023; Yuan et al., 2021). Notably, several existing studies increase the amount of data available to attackers (Mahmood et al., 2021a) or generate synthetic data (Zhou et al., 2020), significantly advancing the development of transfer attacks. Meanwhile, Mahmood et al. (2021b) improve the transferability and robustness of Vision Transformers to adversarial examples

## B DERIVATION OF THE MAXIMUM ENTROPY DISTRIBUTION

The aim of this section is to derive the probability distribution $p_i$ that maximizes entropy under specific constraints. This derivation follows from the Maximum Entropy Principle, which asserts that, given incomplete information, the probability distribution that best represents the current state of knowledge is the one with the maximum entropy.

### B.1 DEFINITION OF ENTROPY

The Shannon entropy for a discrete probability distribution is defined as:

$$S(p) = -\sum_i p_i \log p_i \tag{7}$$

where $p_i$ represents the probability of state $i$, subject to the constraint that the probabilities sum to one:

$$\sum_i p_i = 1 \tag{8}$$

## C THEOREM AND PROOF

**Theorem 1.** *For a discrete random variable $X$ with $k$ possible outcomes, the entropy $H(X)$ is maximized when $X$ follows a uniform distribution.*

*Proof.* Let $X$ be a discrete random variable with probability distribution $P = \{p_1, p_2, \ldots, p_k\}$, where the entropy $H(X)$ is defined as

$$H(X) = -\sum_{i=1}^{k} p_i \log p_i. \tag{9}$$

Our objective is to find the distribution $P$ that maximizes $H(X)$, subject to the constraints that $\sum_{i=1}^{k} p_i = 1$ and $p_i \geq 0$ for all $i$.

We apply the method of Lagrange multipliers, constructing the function

$$\mathcal{L} = -\sum_{i=1}^{k} p_i \log p_i + \lambda \left( \sum_{i=1}^{k} p_i - 1 \right), \tag{10}$$

where $\lambda$ is a Lagrange multiplier. Taking the partial derivative of $\mathcal{L}$ with respect to each $p_i$ and setting it to zero yields

$$\frac{\partial \mathcal{L}}{\partial p_i} = -(\log p_i + 1) + \lambda = 0. \tag{11}$$

Solving this equation, we find that

$$\log p_i = \lambda - 1, \tag{12}$$

which implies that all $p_i$ are equal.

Using the normalization constraint $\sum_{i=1}^{k} p_i = 1$, we deduce that $p_i = \frac{1}{k}$ for all $i$. Thus, the entropy $H(X)$ is maximized when $X$ follows a uniform distribution. □

Therefore, we apply the clustering process for a limited time, and the clustering function is selected based on the results that most closely approximate a uniform distribution. This implies that the number of texts in each cluster is close to $\frac{n}{k}$, where $n$ is the total number of auxiliary texts and $k$ is the number of clusters.

## D  SUBSTITUTE MODEL

### D.1  SUBSTITUTE MODEL ARCHITECTURE

Our substitute model comprises 12 transformer blocks, each with 768 hidden units and 12 self-attention heads. Each transformer block consists of the following substructures:

- **Self-Attention Layer:** The hidden size of the self-attention layer is 768.

- **Position-wise Feed-Forward Network:** The network first projects the output of the attention layer to a 3072-dimensional space using a fully connected layer, followed by a ReLU activation for non-linearity, and finally projects the 3072-dimensional space back to a 768-dimensional space via another fully connected layer.

- **Layer Normalization and Residual Connection:**
  - **Layer Normalization:** Applied to the output of each sub-layer to stabilize training.
  - **Residual Connection:** Adds the normalized output to the input of the sub-layer.

### D.2  SUBSTITUTE MODEL TRAINING

We provide a detailed description of the training of the substitute model with the transformer-based architecture. This substitute model consists of 12 hidden layers with a dimensionality of 768. The activation function "GELU" is used, The dropout rate is 0.4. The training process is optimized with the AdamW optimizer (Yao et al., 2021), with batch size set to 64 and learning rate set to $6e - 3$, over 5 epochs.

## E  COMPUTATION OVERHEAD OF THE SUBSTITUTE MODEL TRAINING

We train five substitute models on a server equipped with a 24 GB NVIDIA 3090 GPU. Each model is trained over two epochs using a dataset containing 100 samples. The training time for a single model is approximately 4 minutes, and the size of each trained model is 418 MB.

## F  DETAILS OF DATA

Table 7: The statistics of datasets.

| Dataset | Train | Test | classes | Labels name |
|---------|-------|------|---------|-------------|
| SST5 | 8544 | 2210 | 5 | Very positive, Positive, Neutral, Negative, Very negative |
| Emotion | 16000 | 2000 | 6 | Sadness, Joy, Love, Anger, Fear, Surprise |

## G  THE URL OF THE VICTIM MODELS

Table 8: The URL of the Victim Models

| Model | Url |
|-------|-----|
| dis-sst5(A) | https://huggingface.co/SetFit/distilbert-base-uncased__sst5__all-train |
| dis-emotion(A) | https://huggingface.co/bhadresh-savani/distilbert-base-uncased-emotion |
| opus-mt(en-zh)(A) | https://huggingface.co/Helsinki-NLP/opus-mt-en-zh |
| ro-sst5(B) | https://huggingface.co/Unso/roberta-large-finetuned-sst5 |
| ro-emotion(B) | https://huggingface.co/SamLowe/roberta-base-go_emotions |
| t5-small(en-fr)(B) | https://huggingface.co/Alexle/T5-small-en-fr |
| Baidu Translate (en-fr) (C) | https://api.fanyi.baidu.com/ |
| Ali Translate (en-zh) (C) | https://translate.alibaba.com/ |

## H DETAILS OF BASELINES

Table 9: The details of the methods employed in the baseline comparisons. The Perturbed Level indicates the target of the attack methods, where "word" denotes the specific words targeted for perturbation, and "char" refers to the characters within a word that are altered by the attack method.

(a) Information on the classification attack method used as the baseline.

| Methods | Perturbed Level | Gradient | Soft-labels | Hard-labels | Knowledge |
|---|---|---|---|---|---|
| Bae | Word | ✗ | ✔ | ✔ | black-box |
| FD | Char | ✔ | ✔ | ✔ | white-box |
| Hotflip | Char | ✔ | ✔ | ✔ | white-box |
| PSO | Word | ✗ | ✔ | ✔ | black-box |
| TextBugger | Char+Word | ✔ | ✔ | ✔ | white-box |
| Leap | Word | ✗ | ✔ | ✔ | black-box |
| CT-GAT | Word | ✗ | ✔ | ✔ | black-box |
| HQA | Word | ✗ | ✔ | ✔ | black-box |
| CEMA | Char+Word | ✗ | ✗ | ✔ | black-box |

(b) Information on the translation attack method used as the baseline.

| Methods | Perturbed Level | Gradient | Soft-labels | Hard-labels | Knowledge |
|---|---|---|---|---|---|
| Hot-trans | Char | ✔ | ✗ | ✗ | white-box |
| kNN | Word | ✔ | ✗ | ✗ | white-box |
| Morphin | Word | ✗ | ✗ | ✔ | black-box |
| RA | Word | ✔ | ✗ | ✗ | white-box |
| Seq2Sick | Word | ✔ | ✗ | ✔ | white-box |
| TransFool | Word | ✗ | ✗ | ✔ | black-box |
| CEMA | Char+Word | ✗ | ✗ | ✔ | black-box |

## I PERFORMANCE EVALUATION ON SIX DOWNSTREAM TASKS

Table 10: The results of six tasks

| | ASR(%) | | | | BLEU | |
|---|---|---|---|---|---|---|
| Data | dis-emotion | ro-emotion | dis-sst5 | ro-sst5 | opus-mt | t5-small |
| SST5 | 75.91 | 74.90 | 67.04 | 62.82 | 0.18 | 0.22 |
| Emotion | 83.25 | 66.85 | 71.35 | 68.40 | 0.17 | 0.27 |

We increase the number of tasks to six downstream tasks, consisting of four classification tasks and two translation tasks, with the corresponding experimental results presented in Table 10. The victim models include dis-emotion, ro-emotion, dis-sst5, ro-sst5, opus-mt, and t5-small. We observe that CEMA achieves an ASR of over 60% on both the SST5 and Emotion datasets, with all BLEU scores below 0.3. These results suggest that the CEMA method can be effectively extended to multi-task learning systems with a broader range of tasks.

## J  PERFORMANCE EVALUATION ON SUMMARY AND TEXT TO IMAGE TASKS

Table 11: The results of translation, summary, and text to image tasks.

| Data | Task | Metric | Score |
|------|------|--------|-------|
| Pokemon | Translation | BLEU | 0.24 |
| | Summary | ROUGE Drop Percentage | 47% |
| | Text to Image | CLIP Drop Percentage | 56% |

We use the Pokemon dataset as the victim text, with the downstream tasks being translation, summarization, and text-to-image generation. The corresponding victim models are t5-small, distilbart-cnn, and Stable Diffusion V2, respectively. BLEU, ROUGE Drop Percentage, and CLIP Drop Percentage are selected as evaluation metrics for the attack. The experimental results, presented in Table 11, indicate that CEMA demonstrates effective attack performance across all tasks, including translation, summarization, and text-to-image generation. These results suggest that CEMA can be effectively extended to other tasks.

## K  EVALUATION ON FEW-SHOT LEARNING AND ADDITIONAL MODEL QUERIES

Influenced by these works (Mahmood et al., 2021a; **?**), we investigate the attack performance of CEMA under both higher and lower query counts in this section. We set the number of queries and the amount of available training data to 10, 50, 100, 1000, and 2000, respectively. The experimental results are presented in Table 12. Our findings indicate that the attack effectiveness increases with the number of queries. Notably, even with just 10 queries, CEMA achieves an attack success rate exceeding 30%.

Table 12: The results of few-shot and additional queries

| Model | Victim Model A | | | | | | Victim Model B | | | | | |
|-------|---------|-------------|---------|---------|-------------|---------|---------|------------|----------|---------|------------|----------|
| | dis-sst5 | dis-emotion | opus-mt | dis-sst5 | dis-emotion | opus-mt | ro-sst5 | ro-emotion | t5-small | ro-sst5 | ro-emotion | t5-small |
| Data | SST5 | | | Emotion | | | SST5 | | | Emotion | | |
| Shot-Size | ASR(%)↑ | ASR(%)↑ | BLEU↓ | ASR(%)↑ | ASR(%)↑ | BLEU↓ | ASR(%)↑ | ASR(%)↑ | BLEU↓ | ASR(%)↑ | ASR(%)↑ | BLEU↓ |
| 2000 | 87.56 | 83.27 | 0.1 | 91.7 | 81.45 | 0.1 | 86.46 | 78.47 | 0.15 | 67.45 | 69.55 | 0.16 |
| 1000 | 83.04 | 76.76 | 0.11 | 88.25 | 76.15 | 0.12 | 84.16 | 73.49 | 0.16 | 66.35 | 67.05 | 0.17 |
| **100** | 73.57 | 62.27 | 0.14 | 80.8 | 65.4 | 0.15 | 75.66 | 64.01 | 0.18 | 60.4 | 59.6 | 0.23 |
| 50 | 63.71 | 45.64 | 0.15 | 71.05 | 53.55 | 0.18 | 71.69 | 59.38 | 0.19 | 58.65 | 57.9 | 0.24 |
| 10 | 38.38 | 32.06 | 0.19 | 43.35 | 37.7 | 0.21 | 59.28 | 46.51 | 0.21 | 46.15 | 41.75 | 0.27 |

## L  REVISIT THE TRANSFER ATTACK

Transfer attacks involve an attacker generating adversarial examples using a substitute model, which are then successfully applied to attack multiple target models. These target models may differ in architecture or training data from the substitute model. This type of attack exploits the shared characteristics of adversarial examples across models, allowing these samples to transfer and affect multiple models. The success of a transfer attack typically depends on the degree of similarity between the substitute model and the target model. Consequently, even if the attacker cannot access the internal information of the target model, they can still use adversarial examples generated from the substitute model to successfully attack the target model.

## M  EVALUATION WITH SIM AND TOTAL-QUERY METRICS

Table 13: Experiment results with similarity and total-query metrics for models A and B

| Dataset | SST5 | | | | Emotion | | | |
|---|---|---|---|---|---|---|---|---|
| Victim Model | Victim Model A | | Victim Model B | | Victim Model A | | Victim Model B | |
| Text Classification | dis-sst5 (A) | | ro-sst5 (B) | | dis-sst5 (A) | | ro-sst5 (B) | |
| Metric | Sim↑ | Total_Qry↓ | Sim↑ | Total_Qry↓ | Sim↑ | Total_Qry↓ | Sim↑ | Total_Qry↓ |
| Bae | 0.888 | 47360 | 0.887 | 47471 | 0.925 | 59626 | 0.924 | 55935 |
| FD | 0.939 | 27758 | 0.982 | 21459 | 0.948 | 66035 | 0.979 | 26719 |
| Hotflip | 0.951 | 25459 | 0.951 | 25945 | 0.942 | 21658 | 0.952 | 22409 |
| PSO | 0.954 | 24398 | 0.954 | 27360 | 0.945 | 19713 | 0.964 | 19757 |
| TextBugger | 0.978 | 69527 | 0.978 | 67007 | 0.981 | 25216 | 0.981 | 25128 |
| Leap | 0.953 | 21548 | 0.944 | 21083 | 0.934 | 15492 | 0.939 | 15315 |
| CT-GAT | 0.939 | 46233 | 0.926 | 82963 | 0.916 | 47338 | 0.927 | 47139 |
| HQA | 0.936 | 64864 | 0.929 | 64267 | 0.934 | 65725 | 0.925 | 47449 |
| **CEMA** | 0.934 | 100 | 0.927 | 100 | 0.926 | 100 | 0.931 | 100 |
| Text Classification | dis-emotion (A) | | ro-emotion (B) | | dis-emotion (A) | | ro-emotion (B) | |
| Metric | Sim↑ | Total_Qry↓ | Sim↑ | Total_Qry↓ | Sim↑ | Total_Qry↓ | Sim↑ | Total_Qry↓ |
| Bae | 0.894 | 60399 | 0.896 | 62013 | 0.926 | 48266 | 0.923 | 48244 |
| FD | 0.921 | 64576 | 0.934 | 36553 | 0.932 | 28310 | 0.982 | 40730 |
| Hotflip | 0.943 | 24001 | 0.946 | 27139 | 0.949 | 31559 | 0.949 | 31824 |
| PSO | 0.968 | 19934 | 0.940 | 20885 | 0.952 | 26144 | 0.951 | 26741 |
| TextBugger | 0.972 | 25084 | 0.986 | 25084 | 0.978 | 67007 | 0.978 | 67029 |
| Leap | 0.968 | 15315 | 0.947 | 15492 | 0.926 | 21503 | 0.911 | 21614 |
| CT-GAT | 0.927 | 47206 | 0.924 | 47493 | 0.904 | 57593 | 0.906 | 56001 |
| HQA | 0.945 | 47382 | 0.931 | 65062 | 0.912 | 49592 | 0.911 | 51184 |
| **CEMA** | 0.934 | 100 | 0.927 | 100 | 0.926 | 100 | 0.931 | 100 |
| Text Translation | opus-mt(en-zh) (A) | | t5-small(en-fr) (B) | | opus-mt(en-zh) (A) | | t5-small(en-fr) (B) | |
| Metric | Sim↑ | Total_Qry↓ | Sim↑ | Total_Qry↓ | Sim↑ | Total_Qry↓ | Sim↑ | Total_Qry↓ |
| Hot-trans | 0.846 | 21570 | 0.842 | 20885 | 0.859 | 20686 | 0.854 | 21680 |
| KNN | 0.873 | 13680 | 0.883 | 13680 | 0.935 | 29481 | 0.906 | 13437 |
| Morphin | 0.894 | 15006 | 0.907 | 24531 | 0.869 | 11183 | 0.887 | 8486 |
| RA | 0.872 | 7028 | 0.865 | 9415 | 0.852 | 6166 | 0.865 | 4663 |
| Seq2sick | 0.881 | 9835 | 0.926 | 13371 | 0.945 | 15669 | 0.892 | 8951 |
| TransFool | 0.949 | 7337 | 0.894 | 8641 | 0.962 | 8597 | 0.924 | 7912 |
| **CEMA** | 0.934 | 100 | 0.927 | 100 | 0.926 | 100 | 0.931 | 100 |

Table 14: Experiment results with similarity and total-query metrics for models A and B

| Data | Victim Model C | Baidu Translate (en-fr) (C) | | Ali Translate (en-zh) (C) | |
|---|---|---|---|---|---|
| | Methods | Sim↑ | Total_Qry↓ | Sim↑ | Total_Qry↓ |
| SST5 | Morphin | 0.904 | 89461 | 0.931 | 107075 |
| | TransFool | 0.921 | 52001 | 0.928 | 68952 |
| | **CEMA** | 0.934 | 100 | 0.934 | 100 |
| Emotion | Morphin | 0.897 | 61416 | 0.915 | 28067 |
| | TransFool | 0.903 | 28067 | 0.923 | 68311 |
| | **CEMA** | 0.931 | 100 | 0.931 | 100 |

We introduce two novel evaluation metrics: the similarity between adversarial examples and the original text, and the number of queries to the victim model. Specifically, *sim* represents the similarity between an adversarial example and the original text, while *Total_Qry* indicates the number of queries made to the victim model. The results are presented in Tables 13 and 14. Our results show that CEMA does not achieve state-of-the-art (SOTA) performance in every scenario. However, in those scenarios where it does not reach SOTA, the similarity remains high. Furthermore, CEMA performs attacks in a black-box setting, where only 100 queries to the victim model are allowed. Given these

stringent attack conditions, we argue that a slight sacrifice in similarity is acceptable in exchange for achieving SOTA performance in the ASR, BLEU, and Query metrics.

## N  THE RESULTS WITH MORE TEXT CLASSIFICATION BASELINES

We incorporate additional methods for adversarial attacks in text classification, such as FGPM, Genetic, and PWWS. The experimental results are presented in Table 15. Compared to these three methods, CEMA also achieves state-of-the-art (SOTA) attack results.

Table 15: The Results of more text classification attack methods

| Data | Method | dis-sst5 | | ro-sst5 | | dis-emotion | | ro-emotion | |
|------|--------|----------|-------|---------|-------|-------------|-------|------------|-------|
| | | ASR(%)↑ | Query | ASR(%)↑ | Query | ASR(%)↑ | Query | ASR(%)↑ | Query |
| SST5 | FGPM | 30.57 | 29.31 | 29.52 | 30.95 | 36.49 | 31.56 | 30.56 | 30.54 |
| | Genetic | 38.46 | 20.35 | 32.31 | 17.50 | 28.53 | 16.49 | 23.58 | 24.24 |
| | PWWS | 33.49 | 18.93 | 31.27 | 23.17 | 34.53 | 21.30 | 42.94 | 28.36 |
| | **CEMA** | **73.57** | **0.045** | **75.66** | **0.045** | **62.27** | **0.045** | **64.01** | **0.045** |
| Emotion | FGPM | 30.57 | 29.31 | 29.52 | 30.95 | 36.49 | 31.56 | 30.56 | 30.54 |
| | Genetic | 38.46 | 20.35 | 32.31 | 17.50 | 28.53 | 16.49 | 23.58 | 24.24 |
| | PWWS | 33.49 | 18.93 | 31.27 | 23.17 | 34.53 | 21.30 | 42.94 | 28.36 |
| | **CEMA** | **80.80** | **0.05** | **60.40** | **0.05** | **65.40** | **0.05** | **59.60** | **0.05** |

## O  PERFORMANCE WITH TRANSFER ATTACKS

Table 16: The results of transfer attack and CEMA

| Dataset | SST5 | | Emotion | |
|---------|------|---|---------|---|
| Victim Model | Victim Model A | Victim Model B | Victim Model A | Victim Model B |
| Text Classification | dis-sst5 | ro-sst5 | dis-sst5 | ro-sst5 |
| Metric | ASR(%)↑ | ASR(%)↑ | ASR(%)↑ | ASR(%)↑ |
| Bae | 29.73 | 26.73 | 17.65 | 16.7 |
| FD | 8.48 | 12.07 | 12.05 | 9.5 |
| Hotflip | 16.66 | 12.31 | 14.85 | 15.65 |
| PSO | 21.88 | 19.16 | 15.55 | 15.9 |
| TextBugger | 19.86 | 12.47 | 22.4 | 8.5 |
| Leap | 14.57 | 27.23 | 15.5 | 9.65 |
| CT-GAT | 12.33 | 17.16 | 9.65 | 13.65 |
| HQA | 24.70 | 16.50 | 13.85 | 15.25 |
| **CEMA** | **73.57** | **75.66** | **80.8** | **64.4** |
| Text Classification | dis-emotion | ro-emotion | dis-emotion | ro-emotion |
| Metric | ASR(%)↑ | ASR(%)↑ | ASR(%)↑ | ASR(%)↑ |
| Bae | 29.49 | 5.77 | 16.75 | 12.05 |
| FD | 19.56 | 68.6 | 9.15 | 7.5 |
| Hotflip | 19.11 | 14.84 | 10.8 | 10.65 |
| PSO | 29.86 | 24.34 | 19.9 | 18.55 |
| TextBugger | 16.89 | 22.89 | 10.95 | 10.3 |
| Leap | 16.5 | 14.11 | 22.05 | 10.85 |
| CT-GAT | 19.92 | 24.15 | 13.85 | 7.8 |
| HQA | 17.88 | 18.44 | 18.46 | 17.7 |
| **CEMA** | **62.27** | **64.01** | **65.40** | **59.6** |
| Text Translation | opus-mt | t5-samll | opus-mt | t5-samll |
| Metric | BLEU↓ | BLEU↓ | BLEU↓ | BLEU↓ |
| Hot-trans | 0.32 | 0.35 | 0.36 | 0.33 |
| KNN | 0.43 | 0.40 | 0.81 | 0.44 |
| Morphin | 0.46 | 0.50 | 0.39 | 0.42 |
| RA | 0.40 | 0.32 | 0.56 | 0.47 |
| Seq2scik | 0.50 | 0.57 | 0.87 | 0.53 |
| TransFoll | 0.94 | 0.58 | 0.93 | 0.87 |
| **CEMA** | **0.14** | **0.18** | **0.15** | **0.23** |

We use the `sst5-setfit-model` and `bert-emotion` models as substitute models for the SST5 and Emotion datasets, respectively. The model URLs for `bert-sst5` and `bert-emotion` are https://huggingface.co/addy88/sst5-setfit-model and https://huggingface.co/bhadresh-savani/bert-base-uncased-emotion, respectively. Additionally, we select `t5-base` as the substitute model for the translation task, with the model available at https://huggingface.co/google-t5/t5-base. We apply various attack algorithms to generate adversarial samples for each model, and then use these adversarial samples to attack the target model. The experimental results, shown in Table 16, indicate that CEMA achieves state-of-the-art (SOTA) attack results compared to transfer attacks.

## P  DEFENSE METHOD

We initiate an extensive exploration of defensive strategies to counter CEMA. In practical systems, we thoroughly investigate various defense mechanisms, including train-free adjustments(Preceding Language Modifier) and adversarial training.

### P.1  PRECEDING LANGUAGE MODIFIER

The victim models used in our study are after-trained models sourced from the Huggingface website, Ali Translator, and Baidu Translator. Since the training details of these pre-trained models are not publicly available, re-training them using adversarial training is infeasible. Consequently, we adopt training-free defense methods. Specifically, we implement the same approach proposed by (Wang et al., 2023) and apply prompt learning techniques to large language models (LLMs) to mitigate adversarial text inputs. For this, we provide CoEdIT-XXL (a LLM used for correcting text errors). The prompt is as follows: "Please revise the text for grammatical errors, improve the spelling, grammar, clarity, concision, and overall readability." The results are presented in Table 20.

"w/o" indicates the absence of a defense method, whereas "w" denotes the use of the CoEdIT-XXL model as a modifier for defense. Even after applying defense mechanisms using large language models, CEMA's attack effectiveness decreases but still maintains a significant level of performance.

Table 17: The results of Preceding Language Modifier

| Victim Model | Dataset | | Metric | w/o | w |
|---|---|---|---|---|---|
| Victim A | SST5 | dis-sst5 | ASR(%)↑ | 73.57 | 40.52 |
| | | dis-emotion | ASR(%)↑ | 62.27 | 36.38 |
| | | opus-mt | BLEU↓ | 0.14 | 0.38 |
| | Emotion | dis-sst5 | ASR(%)↑ | 80.80 | 32.75 |
| | | dis-emotion | ASR(%)↑ | 65.40 | 30.41 |
| | | opus-mt | BLEU↓ | 0.15 | 0.41 |
| Victim B | SST5 | ro-sst5 | ASR(%)↑ | 75.66 | 27.62 |
| | | ro-emotion | ASR(%)↑ | 64.01 | 28.80 |
| | | t5-small | BLEU↓ | 0.18 | 0.24 |
| | Emotion | ro-sst5 | ASR(%)↑ | 60.40 | 31.15 |
| | | ro-emotion | ASR(%)↑ | 59.60 | 33.25 |
| | | t5-small | BLEU↓ | 0.23 | 0.53 |
| Victim C | SST5 | Baidu Translate | BLEU↓ | 0.29 | 0.57 |
| | | Ali Translate | BLEU↓ | 0.15 | 0.49 |
| | Emotion | Baidu Translate | BLEU↓ | 0.35 | 0.72 |
| | | Ali Translate | BLEU↓ | 0.29 | 0.53 |

### P.2  ADVERSARIAL TRAINING

We train four classification models as victim models and conduct adversarial training to evaluate the impact of adversarial training on CEMA's attack effectiveness. All four models are based on the BERT architecture and are labeled Bert1, Bert2, Bert3, and Bert4. Specifically, Bert1 and Bert3 are trained on the SST5 dataset, while Bert2 and Bert4 are trained on the Emotion dataset. The results are presented in Table 18. "w/o" indicates the absence of adversarial training, while "w" represents

the application of adversarial training. Although adversarial training reduces attack effectiveness, CEMA still demonstrates considerable performance.

Table 18: The results of Preceding Language Modifier

| Data | Model | W/O | W |
|---|---|---|---|
| SST5 | Bert1 | 76.27 | 31.31 |
| | Bert2 | 79.81 | 28.94 |
| Emotion | Bert3 | 76.35 | 35.65 |
| | Bert4 | 71.50 | 26.15 |

## Q    THE RESULTS OF RANDOM SHUFFLE

We employ DWB and TextFooler as attack methods for CEMA, allowing them to shuffle between two models during querying and attack phases (Mahmood et al., 2021b). The experimental results, presented in Table 19, demonstrate that Random Shuffle reduces CEMA's attack effectiveness. Nevertheless, CEMA still maintains a reasonably effective level of attack performance under the Random Shuffle defense.

Table 19: The Results of Random Shuffle

| Victim Model | Dataset | | Metric | w/o | w |
|---|---|---|---|---|---|
| Victim A | SST5 | dis-sst5 | ASR(%)↑ | 50.58 | 21.53 |
| | | dis-emotion | ASR(%)↑ | 43.02 | 21.61 |
| | | opus-mt | BLEU↓ | 0.17 | 0.28 |
| | Emotion | dis-sst5 | ASR(%)↑ | 68.45 | 37.50 |
| | | dis-emotion | ASR(%)↑ | 40.25 | 23.20 |
| | | opus-mt | BLEU↓ | 0.19 | 0.36 |
| Victim B | SST5 | ro-sst5 | ASR(%)↑ | 58.67 | 32.67 |
| | | ro-emotion | ASR(%)↑ | 55.32 | 35.64 |
| | | t5-small | BLEU↓ | 0.23 | 0.44 |
| | Emotion | ro-sst5 | ASR(%)↑ | 55.10 | 39.15 |
| | | ro-emotion | ASR(%)↑ | 41.40 | 21.85 |
| | | t5-small | BLEU↓ | 0.28 | 0.51 |

## R    DEFINITION OF TEXT CLASSIFICATION ADVERSARIAL EXAMPLES AND NMT ADVERSARIAL EXAMPLES

### R.1    DEFINITION OF NMT ADVERSARIAL EXAMPLES

We define the source language space as $\mathcal{X}$ and the target language space as $\mathcal{Y}$, examining two NMT systems: the source-to-target model $M_{x \to y}$, which maps $\mathcal{X}$ to $\mathcal{Y}$ to maximize $P(y_{\text{ref}} \mid x)$, and the target-to-source model $M_{y \to x}$, which performs the reverse mapping. After training, these models can reconstruct original sentences as $\hat{x} = g(f(x))$. We propose black-box adversarial testing for NMT using auxiliary data by selecting test sentences from $\mathcal{T} \subset \mathcal{X}$ and generating adversarial cases $\delta \in \Delta$ to perturb inputs $x' = x + \delta$ such that $f(x')$ diverges significantly from $f(x)$.

**NMT Adversarial Example:** An NMT adversarial example is a sentence in

$$\mathcal{A} = \{x' \in \mathcal{X} \mid \exists x \in \mathcal{T}\},$$
$$here \, \|x' - x\| < \epsilon \wedge S_t\left(y, y_{\text{ref}}\right) \geq \gamma \wedge S_t\left(y', y_{\text{ref}}\right) < \gamma' \tag{13}$$

where function $f$ represents the NMT model. The variables $x$ and $x'$ represent the original text and the adversarial test case, respectively, while $y$ and $y'$ stand for their respective translations. In detail, $y = f(x)$ and $y' = f(x')$. The function $S_t(\cdot, \cdot)$ gauges the similarity between two sentences. Additionally, $\gamma$ and $\gamma'$ denote thresholds for acceptable translation quality. Translation quality is deemed unacceptable if $\gamma'$ drops below $\gamma$.

## R.2 Definition of Text Classification Adversarial Examples

**Definition of Text Classification Adversarial Examples:** Let $X = \{x_1, x_2, \ldots, x_n\}$ denote a set of text inputs, where each $x_i$ is a text document (e.g., sentence or paragraph). Let $f(\cdot)$ represent a text classification model, where:

$$f : X \to Y$$

is a mapping from the input space $X$ to the label space $Y$, with $Y = \{y_1, y_2, \ldots, y_m\}$ representing the set of possible class labels (e.g., positive, negative, neutral).

Given an input $x \in X$ and its corresponding true label $y_{\text{true}} = f(x)$, an *adversarial example* $\hat{x}$ is a perturbed version of the input $x$ that is intentionally crafted to cause the model to misclassify it, while remaining perceptually and semantically similar to the original text. Formally, an adversarial example is defined as:

$$\hat{x} = x + \delta$$

where $\delta$ is a small perturbation that satisfies:

$$\|\delta\| \leq \epsilon$$

Here, $\|\delta\|$ represents the magnitude of the perturbation (e.g., measured in terms of the number of word substitutions or sentence modifications), and $\epsilon$ is a threshold that bounds the maximum allowable perturbation.

Additionally, we impose a *semantic similarity* constraint, ensuring that the perturbation $\delta$ does not alter the meaning of the input significantly. This is formalized as:

$$\text{Sim}(x, \hat{x}) \leq \gamma$$

where $\text{Sim}(x, \hat{x})$ denotes a semantic similarity measure (such as cosine similarity) between the original input $x$ and the adversarial example $\hat{x}$, and $\gamma$ is a predefined threshold that controls the acceptable level of semantic similarity. This ensures that the adversarial example $\hat{x}$ remains semantically close to $x$, while still leading to a misclassification.

The adversarial example $\hat{x}$ causes the model to output a different class than the true label:

$$f(\hat{x}) \neq y_{\text{true}} \quad \text{and} \quad f(x) = y_{\text{true}}$$

$$\hat{x} = \text{argmin}_{x' \in X} \mathcal{L}(f(x'), y_{\text{true}}) \quad \text{subject to} \quad \|x' - x\| \leq \epsilon \quad \text{and} \quad \text{Sim}(x, x') \geq \gamma$$

where $\mathcal{L}(\cdot)$ is the loss function used to measure the discrepancy between the predicted label $f(x')$ and the true label $y_{\text{true}}$.

## S  THE EXPERIMENT FOR VERIFYING INDEPENDENCE

Table 20: The experimental results for verifying independence.

| Method A | Method B | P(A) | P(B) | P(A)*P(B) | P(AB) | P(A)*P(B)-P(AB) |
|---|---|---|---|---|---|---|
| DWB | FD | 52.50% | 40.50% | 21.26% | 19.00% | 2.26% |
| DWB | Textbugger | 52.50% | 72.50% | 38.06% | 35.50% | 2.56% |
| DWB | Hotflip | 52.50% | 72.50% | 38.06% | 37.00% | 1.06% |
| DWB | PSO | 52.50% | 76.50% | 40.16% | 35.00% | 5.16% |
| FD | Textbugger | 40.50% | 72.50% | 29.36% | 31.00% | -1.64% |
| FD | Hotflip | 40.50% | 72.50% | 29.36% | 31.50% | -2.14% |
| FD | PSO | 40.50% | 76.50% | 30.98% | 30.50% | 0.48% |
| Textbugger | Hotflip | 72.50% | 72.50% | 52.56% | 57.50% | -4.94% |
| Textbugger | PSO | 72.50% | 76.50% | 55.46% | 54.00% | 1.46% |
| Hotflip | PSO | 72.50% | 76.50% | 55.46% | 58.00% | -2.54% |
| Average | | | | 39.07% | 38.90% | 0.17% |

We employ the DWB, FD, TextBugger, Hotflip, and PSO methods to generate adversarial examples. Since the exact success probabilities of each method's attacks are unavailable, we estimate these probabilities based on the observed frequency of successful attacks. In the table, we report the frequency $P(AB)$ of both methods successfully attacking, as well as the individual success frequencies $P(A)$ for Method A and $P(B)$ for Method B. Our findings indicate that $P(AB)$ closely approximates $P(A) \times P(B)$, with the average deviation $P(A) \times P(B) - P(AB)$ being just 0.17%. The detailed experimental results are provided in Table 20. Event independence is defined as the occurrence of event A having no effect on the occurrence of event B. Therefore, we assume that the success of adversarial examples generated by Method A does not influence the success of those generated by Method B.

## T  SUPPLEMENTARY EXPLANATION FOR THE NON-INDEPENDENT CASE IN SECTION CANDIDATE ADVERSARIAL EXAMPLE GENERATION

The probability of successfully attacking the victim model using adversarial examples generated by methods 1, 2, ..., $n$ is greater than or equal to the probability of successfully attacking the victim model using adversarial examples generated by method 1 alone. This is because, when only method 1 is used, there is only one candidate adversarial example per victim text. In contrast, when $n$ methods are employed, there are $n$ candidate adversarial examples for each victim text, including the one generated by method 1. Therefore, the probability of successfully attacking the victim model using adversarial examples generated by $n$ methods is greater than or equal to the probability of successfully attacking the victim model using adversarial examples from method 1 alone. The probabilities are equal only when method 1 achieves the maximum success rate for all victim texts. However, the SST5 and Emotion datasets contain 2,210 and 2,000 victim texts, respectively, making it unlikely that method 1 will achieve the maximum success rate across all victim texts. Thus, we conclude that, in most cases, the probability of successfully attacking the victim model using adversarial examples generated by $n$ methods is greater than when using adversarial examples generated by method 1 alone.

Furthermore, based on this property, we can deduce that, in most cases, the probability of successfully attacking the victim model using adversarial examples generated by methods 1, 2, ..., $n$ is greater than when using adversarial examples generated by methods 1, 2, ..., $m$, where $n > m$. In other words, employing more methods to generate adversarial examples increases the likelihood of a successful attack on the victim model.

## U  MULTI-MODEL MULTI-TASK LEARNING (MMMTL)

Multi-model Multi-task Learning (MMMTL) is a machine learning method that combines multiple learning models with multiple tasks. It is a combination of Multi-task Learning (MTL) and Multi-model Learning, aiming to improve model performance by jointly optimizing multiple tasks, especially when dealing with multiple related tasks.

## U.1 KEY CONCEPTS

### U.1.1 MULTI-TASK LEARNING (MTL)

In traditional machine learning, each model typically handles a single task. In contrast, Multi-task Learning (MTL) involves jointly training multiple related tasks with a shared model. The goal is to allow the model to simultaneously optimize multiple objectives by sharing representations, knowledge, or parameters. Common applications include sentiment analysis and text classification, where the same features can be used for multiple tasks (e.g., predicting sentiment labels and classifying news articles). For instance, training a neural network to simultaneously perform two tasks: image classification and object detection.

### U.1.2 MULTI-MODEL LEARNING

Unlike traditional single-model approaches, Multi-model Learning uses multiple independent or combined models to solve a problem. Each model may focus on different aspects of the problem or apply different algorithms to address the same task. For example, using multiple models such as neural networks, decision trees, and support vector machines to handle the same task, thereby leveraging the strengths of each model.

### U.1.3 MULTI-MODEL MULTI-TASK LEARNING (MMMTL)

MMMTL is a method that combines Multi-task Learning and Multi-model Learning. The core idea is to use multiple models (e.g., neural networks, decision trees, support vector machines, etc.) to learn multiple related tasks, with these models sharing some information or parameters. This means that during training, MMMTL models handle multiple tasks and models simultaneously, enabling each model to learn across multiple tasks while sharing representations and knowledge between tasks.

