objective of our attack is to compromise the performance of all tasks within a multi-task model. Adversarial examples are designed to generate inputs that universally degrade the model's performance across all tasks, rather than targeting a single task. This means the attacker seeks to create an input that not only disrupts one specific task but also negatively impacts the performance of other tasks within the model. In the context of multitask learning, specifically for text classification, the objective is to ensure that the original text and the adversarial example produce different output labels in the target model, *i.e.* $y_{\text{adv}} \neq y_{\text{ori}}$. For the translation task within multi-task learning, the goal is to ensure that the original text and the adversarial example lead to a significant semantic divergence in the generated translations, *i.e.* $\arg\min \text{BLEU}(y_{\text{adv}}, y_{\text{

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

 exhibits only slight variations between different clustering methods. These changes remain minimal. In contrast, in the translation task, the BLEU score fluctuates more significantly depending on the clustering method used. Although these fluctuations are more pronounced, no consistent pattern emerges. No single clustering method consistently achieves SOTA attack performance across all scenarios. The average ASR for the Spectral, KMeans, and BIRCH clustering methods is 67.71%, 65.21%, and 65.05%, respectively, while the corresponding average BLEU scores are 0.21, 0.20, and 0.21. Therefore, we conclude that *while clustering methods do influence attack performance, their impact is largely random and does not consistently favor one method over another.*

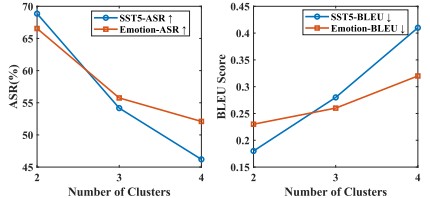

Figure 3: The average ASR and BLUE of different numbers of clusters. Fewer clusters result in better attack performance.

## 5.6 THE IMPACT OF VECTORIZATION METHODS

Given that our multi-task framework includes a translation task, we utilize the multilingual pre-trained model mT5 (Xue, 2020) for text vectorization. Additionally, we employ the XLM-R (Conneau, 2019) pre-trained model and the one-hot (Rodríguez et al., 2018) encoding method. One-hot encoding

Table 5: Performance of CEMA under various vectorization methods.

| Data | Vectorization Method | Victim Model A | | | Victim Model B | | | Victim Model C | |
|---|---|---|---|---|---|---|---|---|---|
| | | dis-sst5 | dis-emotion | opus-mt (en-zh) | ro-sst5 | ro-emotion | t5-small (en-fr) | Baidu Translate (en-fr) | Ali Translate (en-zh) |
| | | ASR(%)↑ | ASR(%)↑ | BLEU↓ | ASR(%)↑ | ASR(%)↑ | BLEU↓ | BLEU↓ | BLEU↓ |
| SST5 | mT5 | 73.57 | 62.27 | 0.14 | 75.66 | 64.01 | 0.18 | 0.29 | 0.13 |
| | XLM-R | 73.55 | 61.09 | 0.17 | 74.90 | 63.44 | 0.19 | 0.38 | 0.11 |
| | one-hot | 73.57 | 61.24 | 0.11 | 75.09 | 62.90 | 0.13 | 0.23 | 0.15 |
| Emotion | mT5 | 80.80 | 65.40 | 0.15 | 60.40 | 59.60 | 0.23 | 0.35 | 0.21 |
| | XLM-R | 81.05 | 64.95 | 0.19 | 53.80 | 53.75 | 0.19 | 0.37 | 0.16 |
| | one-hot | 81.05 | 65.65 | 0.18 | 62.35 | 59.90 | 0.27 | 0.43 | 0.25 |

Table 6: Zero-shot attack performance of CEMA. In a zero-shot scenario, attackers do not have access to auxiliary data with the same distribution as the victim texts. When the victim texts are SST5 data, attackers only need to recognize that they are sentiment-related, allowing them to collect 100 unlabeled texts from sentiment-related datasets, such as the Emotion dataset, as auxiliary texts.

| Victim Data | Access Data | Victim Model A | | | Victim Model B | | | Victim Model C | |
|---|---|---|---|---|---|---|---|---|---|
| | | dis-sst5 | dis-emotion | opus-mt (en-zh) (A) | ro-sst5 (A) | ro-emotion (A) | t5-small (en-fr) | Baidu Translate (en-fr) | Ali Translate (en-zh) |
| | | ASR(%)↑ | ASR(%)↑ | BLEU↓ | ASR(%)↑ | ASR(%)↑ | BLEU↓ | BLEU↓ | BLEU↓ |
| SST5 | SST5 | **73.57** | **62.27** | 0.14 | **75.66** | **64.01** | 0.18 | **0.29** | **0.15** |
| | Emotion | 64.00 | 60.80 | **0.18** | 59.20 | 52.00 | 0.22 | 0.36 | 0.27 |
| Emotion | Emotion | **80.80** | **65.40** | **0.15** | **60.40** | **59.60** | **0.23** | **0.35** | **0.29** |
| | SST5 | 66.40 | 36.00 | 0.21 | 48.80 | 46.40 | 0.36 | 0.44 | 0.42 |

converts categorical data into binary vectors, with each category represented by a unique vector where one element is set to 1 and all others to 0. To reduce the risk of data leakage, we restrict the use of one-hot encoding to 100 samples from the additional dataset. As shown in Figure 2 and Table 5, different vectorization methods have no significant impact on attack performance in the classification task. In the translation task, while vectorization methods cause fluctuations in attack results, these variations are irregular, and no single method consistently achieves SOTA performance across all datasets and victim models. Specifically, the average ASR for the mT5, XLM-R, and one-hot vectorization methods is 67.71%, 65.81%, and 67.72%, respectively, while the average BLEU scores are 0.21, 0.22, and 0.22, respectively. Therefore, we conclude that *vectorization methods do not substantially influence attack performance.*

## 5.7 ZERO-SHOT ATTACK OF CEMA

In this section, we evaluate CEMA's effectiveness under more stringent conditions, where the attacker can only access data related to the training set. For example, both the SST5 and Emotion datasets are related to sentiment analysis, but their label spaces and distributions differ significantly. To test this, we used 100 unlabeled texts from the Emotion validation set as auxiliary data for the SST5 attack, and vice versa. The experimental results, presented in Table 6, show that even with limited auxiliary data and significant differences between the auxiliary data and victim texts, CEMA achieves an attack success rate of 66.40% and a BLEU score of 0.27. The findings indicate that an attacker requires only partial knowledge of the training dataset and gather the relevant data from the Internet. Utilizing CEMA, they can then execute a substantial attack on the multi-task system.

## 6 CONCLUSION

In this paper, we present a more practical multi-task learning scenario where attackers can only access final black-box outputs through limited queries. To address this challenge, we propose the CEMA method, which achieves state-of-the-art (SOTA) performance in experimental evaluations with just 100 queries and black-box outputs. Furthermore, CEMA can incorporate any text classification attack algorithm, and its performance improves as the number of attack algorithms increases. In the future, we aim to extend CEMA to multi-task models across other modalities.

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

This appendix includes our supplementary materials as follows:

- Related Work in Section A.

- Manifold-Based Analysis of Adversarial Examples's transferability in Section B

- Derivation of the maximum entropy distribution in Section C

- Union Bound Theorem and Detailed Proof in Section D

- More detail of substitute model architecture in Section E

- More detail of Data in Section F

- Url of the data and model used in Section G

- Details of Baselines in Section H

## A  RELATED WORK

### A.1  TEXT CLASSIFICATION ADVERSARIAL ATTACK

In historical textual adversarial research, the predominant methods revolve around scenarios with singular output results (Waghela et al., 2024; Han et al., 2024; Zhu et al., 2024; Kang et al., 2024). These studies focus on the techniques for morphing the original text into adversarial counterparts, including the manipulation of pivotal chars (Ebrahimi et al., 2018b; Gil et al., 2019; Ebrahimi et al., 2018a; Gao et al., 2018; Ren et al., 2019; Jin et al., 2020; Li et al., 2019), words (Wang et al., 2022; Guo et al., 2021; Meng & Wattenhofer, 2020; Sato et al., 2018; Cheng et al., 2019; Lee et al., 2022; Li et al., 2020; Hu et al., 2024; Liu et al., 2024; 2023; Li et al., 2019) and sentence. These methods are segmented into three distinct categories based on the response from the target model, encompassing white-box attacks, soft-label black-box attacks, and hard-label black-box attacks. In white-box attacks, adversaries gain full access to all relevant information about the target model. The Hotflip (Ebrahimi et al., 2018b) sequentially replaces crucial words based on their calculated importance scores. The FD method (Papernot et al., 2016) constructs adversarial examples depending on the model's gradient information. In soft-label black-box attacks, numerous methods are geared towards disturbing the words in accordance with output probabilities (Lee et al., 2022; Maheshwary et al., 2021b; Wang et al., 2021; Li et al., 2020). Bert-ATTACK (Li et al., 2020) focuses on word attacks using a refined Bert model. SememePSO (Zang et al., 2020) enhances the search landscape to construct adversarial examples. Bae (Garg & Ramakrishnan, 2020) is an attack strategy centered on BERT to replace words. Simultaneously, the DeepWordBug (DWB) method (Gao et al., 2018) prioritizes the words for assault based on the output probabilities. Hard-label adversarial attacks present a more realistic scenario. HLGA (Maheshwary et al., 2021a) employs stochastic starting words and employs a genetic algorithm to craft adversarial examples. HQA-attack (Liu et al., 2024) starts by maximally restoring original words, reducing disruption. It then uses synonyms of remaining altered words to enhance the adversarial example.

### A.2  NEURAL MACHINE TRANSLATION ADVERSARIAL ATTACK

Neural Machine Translation (NMT) models, which automatically convert input sentences into translated output, have achieved remarkable results by employing deep neural networks like Transformers (Bahdanau, 2014; Vaswani, 2017). These models are now extensively used across various applications due to their high performance. However, erroneous outputs generated by NMT models can lead to significant risks, particularly in security-sensitive contexts. Recent research has explored adversarial attacks targeting NMT models to address these concerns. Character-level NMT models are highly vulnerable to character manipulations such as typos in a block-box setting (Belinkov & Bisk, 2017; Ebrahimi et al., 2018a). as well as pushing/removing words from the translation. However, character manipulations and typos are easily detected by humans or review strategies. Hence, most adversarial attacks against NLP and NMT systems use a word replacement strategy instead. Seq2sickCheng et al. (2020) proposes a projected gradient method combined with group lasso and gradient regularization, conducting non-overlapping attacks and targeted keyword attacks. Similarly, Transfool (Sadrizadeh et al., 2023) also uses the gradient projection method, defining a new optimization problem and linguistic constraints to compute semantic-preserving and fluent attacks against NMT models. Mor-

phin (Tan et al., 2020) generates plausible and semantically similar adversaries by perturbing the inflections in clean examples to investigate the robustness of NLP models to inflectional perturbation. kNN (Michel et al., 2019)is a white-box untargeted attack against NMT models that substitutes some words with their neighbors in the embedding space. RGZou et al. (2019)investigates the issue by generating adversarial examples through a new paradigm based on reinforcement learning, which generates more reasonable tokens and secures semantic constraints.

### A.3 Mutil-task Adversarial Attack

A Multi-task Adversarial Attack is an adversarial machine learning strategy designed to generate examples that deceive multiple models or systems simultaneously (Guo et al., 2020; Ghamizi et al., 2022), rather than just one. As far as we know, there is currently no related work on multi-task adversarial attacks in the field of text. In other fields, MTA (Guo et al., 2020) is designed to generate adversarial perturbations for all three pre-trained classifiers simultaneously by leveraging shared knowledge among tasks. There is an attack method (Sobh et al., 2021) that targets visual perception in autonomous driving, which is applied in a wide variety of multi-task visual perception deep networks in distance estimation, semantic segmentation, motion detection, and object detection. MTADV (Wang et al., 2024) is a multitask adversarial attack against facial authentication, which is effective against various facial data sets.

## B Transferability of Adversarial Examples: A Manifold-Based Analysis

In this section, we present a rigorous mathematical analysis of the transferability of adversarial examples between a surrogate model and a victim model. Specifically, we analyze a scenario in which adversarial examples are generated on a surrogate model trained with cluster labels obtained through clustering. Despite the dissimilarity between the surrogate and victim models, the adversarial examples exhibit strong transferability. We use manifold theory to provide a deeper understanding of this phenomenon, focusing on the shared geometric properties between the models.

### B.1 The Manifold Hypothesis and Data Geometry

The *manifold hypothesis* posits that high-dimensional data, such as images or text, actually lie on or near a lower-dimensional manifold embedded in the high-dimensional input space. Let $x \in \mathbb{R}^n$ represent a data point in the high-dimensional input space. The hypothesis assumes that $x$ lies on a manifold $\mathcal{M} \subset \mathbb{R}^n$, where $\dim(\mathcal{M}) = d \ll n$. This implies that, although the data exists in a high-dimensional space, its intrinsic dimensionality is much lower, captured by the manifold structure.

Formally, we assume the existence of a differentiable embedding $\phi$ that maps points from a low-dimensional latent space $z \in \mathbb{R}^d$ to the high-dimensional input space $\mathbb{R}^n$:

$$x = \phi(z), \quad z \in \mathbb{R}^d, \quad x \in \mathcal{M} \tag{7}$$

The manifold $\mathcal{M}$ provides a lower-dimensional representation of the data's intrinsic structure. This assumption is central to understanding how adversarial examples exploit local geometries of the data.

### B.2 Tangent Space and Adversarial Perturbations

For each point $x \in \mathcal{M}$, the manifold has a tangent space $T_x\mathcal{M}$, which is a linear approximation of the manifold at $x$. The tangent space can be described as the image of the differential map $D\phi(z)$ at the point $z$:

$$v \in T_x\mathcal{M} \quad \text{iff} \quad \frac{d}{dt}\phi(z+tv)\Big|_{t=0} \in T_x\mathcal{M} \tag{8}$$

In adversarial attacks, the perturbation $\eta \in \mathbb{R}^n$ is added to the input $x$, causing it to move off the manifold or within the neighborhood of $\mathcal{M}$. The goal of adversarial perturbation is to make the

modified sample $x' = x + \eta$ fool the classification model. Typically, we constrain the perturbation $\eta$ to lie within a small ball around $x$, i.e., $\|\eta\| \leq \epsilon$.

Given the manifold structure, the perturbation $\eta$ can be viewed as lying in the tangent space $T_x \mathcal{M}$:

$$x' = x + \eta, \quad \eta \in T_x \mathcal{M} \tag{9}$$

This means that the perturbation $\eta$ primarily affects the local geometry of the data, altering the input within the locally linear approximation of the manifold.

### B.3 OPTIMIZATION OF ADVERSARIAL PERTURBATIONS

The goal of generating adversarial examples is to find a perturbation $\eta$ that maximizes the loss function $L(f(x), y)$, where $f$ is the classification model, $x$ is the input, and $y$ is the true label. The perturbation is constrained by $\|\eta\| \leq \epsilon$, ensuring that the modification to the input is imperceptible.

Mathematically, this problem can be formulated as the following optimization problem:

$$\eta = \arg \max_{\|\eta\| \leq \epsilon} L(f(x + \eta), y) \tag{10}$$

To approximate this solution, we apply a first-order Taylor expansion of the loss function around $x$:

$$L(f(x + \eta), y) \approx L(f(x), y) + \nabla_x L(f(x), y)^T \eta \tag{11}$$

Thus, the adversarial perturbation is chosen to align with the gradient of the loss function $\nabla_x L(f(x), y)$. The optimal perturbation $\eta$ is given by:

$$\eta = \epsilon \cdot \frac{\nabla_x L(f(x), y)}{\|\nabla_x L(f(x), y)\|} \tag{12}$$

Therefore, the adversarial example $x'$ is:

$$x' = x + \epsilon \cdot \frac{\nabla_x L(f(x), y)}{\|\nabla_x L(f(x), y)\|} \tag{13}$$

### B.4 MANIFOLD LEARNING IN SURROGATE MODELS

In transfer-based attacks, a surrogate model $f_{\text{proxy}}$ is often trained on auxiliary data using cluster labels obtained through clustering. Assume that the clustering algorithm divides the data into two clusters corresponding to two pseudo-classes, $\mathcal{M}_1 \subset \mathbb{R}^n$ and $\mathcal{M}_2 \subset \mathbb{R}^n$, representing different regions of the input manifold $\mathcal{M}$. These pseudo-classes are determined based on data similarities (e.g., through a clustering algorithm such as k-means).

Let the cluster labels be denoted by $\hat{y}_i \in \{0, 1\}$, so that:

$$x_i \in \mathcal{M}_1 \quad \text{if} \quad \hat{y}_i = 0, \quad x_i \in \mathcal{M}_2 \quad \text{if} \quad \hat{y}_i = 1 \tag{14}$$

The surrogate model $f_{\text{proxy}}$ is then trained to separate these two clusters by learning a decision boundary between the manifolds $\mathcal{M}_1$ and $\mathcal{M}_2$:

$$f_{\text{proxy}}(x) = \begin{cases} 0, & \text{if } x \in \mathcal{M}_1 \\ 1, & \text{if } x \in \mathcal{M}_2 \end{cases} \tag{15}$$

This model learns the local geometric structure of the auxiliary data manifold and attempts to separate the data based on the clustering-derived labels.

## B.5 VICTIM MODEL'S MANIFOLD REPRESENTATION AND TRANSFERABILITY

The victim model $f_{\text{target}}$ is trained on the same or a similar data distribution. Let the victim model learn two classes corresponding to two regions of the data manifold, $\mathcal{M}_{\text{target},1}$ and $\mathcal{M}_{\text{target},2}$. These manifolds can be expressed as transformations of the original data manifold $\mathcal{M}$ through mappings $g_{\text{target},1}$ and $g_{\text{target},2}$:

$$\mathcal{M}_{\text{target},1} = g_{\text{target},1}(\mathcal{M}), \quad \mathcal{M}_{\text{target},2} = g_{\text{target},2}(\mathcal{M}) \tag{16}$$

Although the class labels between the surrogate and victim models differ, the geometric structure of the underlying data manifold remains largely similar. Therefore, if the decision boundaries learned by the surrogate model in regions $\mathcal{M}_1$ and $\mathcal{M}_2$ coincide with regions of high sensitivity in the victim model, adversarial examples generated on the surrogate model can transfer effectively.

## B.6 GEOMETRIC TRANSFERABILITY OF ADVERSARIAL EXAMPLES

In regions where the geometric properties of the surrogate and victim models are similar, adversarial examples generated on $f_{\text{proxy}}$ can also transfer to $f_{\text{target}}$. Specifically, let the decision boundaries of the surrogate model and victim model be denoted as $\partial\mathcal{M}_{\text{proxy}}$ and $\partial\mathcal{M}_{\text{target}}$, respectively. If these boundaries are geometrically close in some region of the manifold, i.e.,

$$\partial\mathcal{M}_{\text{proxy}} \approx \partial\mathcal{M}_{\text{target}} \quad \text{locally}, \tag{17}$$

then adversarial perturbations that cross $\partial\mathcal{M}_{\text{proxy}}$ are likely to also cross $\partial\mathcal{M}_{\text{target}}$.

## B.7 JACOBIAN MATRICES AND GRADIENT TRANSFER

A key geometric aspect of adversarial transferability is the similarity in the local gradient fields of the surrogate and victim models. This can be measured through the Jacobian matrices of the models, denoted as $J_{\text{proxy}}(x)$ and $J_{\text{target}}(x)$, respectively. In regions where these Jacobian matrices are similar, i.e.,

$$J_{\text{proxy}}(x) \approx J_{\text{target}}(x), \quad \forall x \in \mathcal{M}, \tag{18}$$

the adversarial perturbation $\eta$ that is effective for the surrogate model will also be effective for the victim model, thus enhancing the transferability of adversarial examples.

## B.8 CONCLUSION

Through the detailed mathematical formulas and geometric explanations, we arrive at the following conclusions:

- **Manifold Hypothesis**: Data resides on low-dimensional manifolds, and both the surrogate model and the victim model learn different decision boundaries on these manifolds.

- **Tangent Space Perturbations**: Adversarial examples are generated by perturbing within the tangent space $T_x\mathcal{M}$ of the data manifold, with the perturbation optimized in the direction of the gradient.

- **Shared Geometric Properties**: The surrogate and victim models share local geometric properties of the manifold (e.g., curvature, Jacobian matrices), which leads to the transferability of adversarial examples.

- **Locality of Adversarial Perturbations**: The adversarial perturbation impacts local vulnerable regions in the surrogate model, which often correspond to similar vulnerable regions in the victim model, ensuring successful transfer.

- **Training on cluster labels**: The surrogate model trained with cluster labels derived from clustering learns local geometric structures of the data manifold, and these structures are shared with the victim model, explaining the high transferability of adversarial examples

generated from the surrogate model, even though the global structures of the two models differ.

## C  DERIVATION OF THE MAXIMUM ENTROPY DISTRIBUTION

The aim of this section is to derive the probability distribution $p_i$ that maximizes entropy under specific constraints. This derivation follows from the Maximum Entropy Principle, which asserts that, given incomplete information, the probability distribution that best represents the current state of knowledge is the one with the maximum entropy.

### C.1  DEFINITION OF ENTROPY

The Shannon entropy for a discrete probability distribution is defined as:

$$S(p) = -\sum_i p_i \log p_i \tag{19}$$

where $p_i$ represents the probability of state $i$, subject to the constraint that the probabilities sum to one:

$$\sum_i p_i = 1 \tag{20}$$

### C.2  CONSTRAINTS

In addition to the normalization constraint $\sum_i p_i = 1$, we consider an additional constraint on the expected value of a physical observable $f$, such that its expected value $\langle f \rangle$ is known. This constraint is expressed as:

$$\sum_i p_i f_i = \langle f \rangle \tag{21}$$

Thus, we aim to find a probability distribution $p_i$ that maximizes the entropy $S(p)$, while satisfying both the normalization condition and the expectation constraint.

### C.3  APPLICATION OF LAGRANGE MULTIPLIERS

To incorporate these constraints into the maximization of entropy, we employ the method of Lagrange multipliers. The Lagrange multipliers $\lambda_0$ and $\lambda_1$ correspond to the normalization and expectation constraints, respectively. The Lagrangian is defined as:

$$\mathcal{L}(p_i, \lambda_0, \lambda_1) = -\sum_i p_i \log p_i + \lambda_0 \left( \sum_i p_i - 1 \right) + \lambda_1 \left( \sum_i p_i f_i - \langle f \rangle \right) \tag{22}$$

### C.4  MAXIMIZATION OF THE LAGRANGIAN

To maximize the entropy, we differentiate the Lagrangian with respect to $p_i$, yielding:

$$\frac{\partial \mathcal{L}}{\partial p_i} = -(\log p_i + 1) + \lambda_0 + \lambda_1 f_i \tag{23}$$

Setting this derivative equal to zero to find the extremum, we obtain:

$$-(\log p_i + 1) + \lambda_0 + \lambda_1 f_i = 0 \tag{24}$$

which simplifies to:

$$\log p_i = \lambda_0 - 1 + \lambda_1 f_i \tag{25}$$

Exponentiating both sides yields the general form of the probability distribution:

$$p_i = e^{\lambda_0 - 1 + \lambda_1 f_i} \tag{26}$$

Introducing a constant $A = e^{\lambda_0 - 1}$, this becomes:

$$p_i = A e^{\lambda_1 f_i} \tag{27}$$

### C.5 NORMALIZATION AND DETERMINATION OF $A$

The normalization condition $\sum_i p_i = 1$ allows us to solve for the constant $A$. Substituting $p_i = A e^{\lambda_1 f_i}$ into the normalization condition, we get:

$$A \sum_i e^{\lambda_1 f_i} = 1 \tag{28}$$

Hence, $A$ is given by:

$$A = \frac{1}{\sum_i e^{\lambda_1 f_i}} \tag{29}$$

Thus, the probability distribution that maximizes entropy under the given constraints is:

$$p_i = \frac{e^{\lambda_1 f_i}}{\sum_i e^{\lambda_1 f_i}} \tag{30}$$

### C.6 DETERMINATION OF $\lambda_1$

The Lagrange multiplier $\lambda_1$ is determined using the expectation constraint:

$$\sum_i p_i f_i = \langle f \rangle \tag{31}$$

Substituting $p_i = \frac{e^{\lambda_1 f_i}}{\sum_i e^{\lambda_1 f_i}}$ into this constraint yields:

$$\frac{\sum_i f_i e^{\lambda_1 f_i}}{\sum_i e^{\lambda_1 f_i}} = \langle f \rangle \tag{32}$$

This implicit equation must be solved to determine the value of $\lambda_1$. Typically, this equation requires numerical methods for its solution. The value of $\lambda_1$ ensures that the expectation constraint is satisfied.

### C.7 CONCLUSION

The resulting distribution that maximizes entropy, subject to both normalization and expectation constraints, is:

$$p_i = \frac{e^{\lambda_1 f_i}}{\sum_i e^{\lambda_1 f_i}} \tag{33}$$

where the Lagrange multiplier $\lambda_1$ is determined by the equation:

$$\frac{\sum_i f_i e^{\lambda_1 f_i}}{\sum_i e^{\lambda_1 f_i}} = \langle f \rangle \tag{34}$$

This form of the probability distribution is widely used in statistical mechanics and information theory. For instance, in statistical mechanics, the Boltzmann distribution arises as a specific case of this general result. The maximum entropy principle thus provides a systematic approach to determining the most likely distribution, given incomplete information and known constraints.

## D  UNION BOUND THEOREM AND DETAILED PROOF

The **Union Bound** is a fundamental result in probability theory that gives an upper bound on the probability of the union of several events. Formally, for a given set of events $A_1, A_2, \ldots, A_n$ in a probability space, the Union Bound states:

$$P\left(\bigcup_{i=1}^{n} A_i\right) \leq \sum_{i=1}^{n} P(A_i) \tag{35}$$

### D.1  PROOF

We will prove this statement using induction and basic properties of probability theory, such as additivity and monotonicity. We break the proof into several key steps for clarity.

#### D.1.1  STEP 1: BASE CASE FOR TWO EVENTS

We begin by proving the Union Bound for two events $A_1$ and $A_2$. Using the inclusion-exclusion principle, we know:

$$P(A_1 \cup A_2) = P(A_1) + P(A_2) - P(A_1 \cap A_2) \tag{36}$$

Since probabilities are non-negative, we know that:

$$P(A_1 \cap A_2) \geq 0 \tag{37}$$

Thus, we have:

$$P(A_1 \cup A_2) \leq P(A_1) + P(A_2) \tag{38}$$

This inequality establishes the Union Bound for two events. We now extend this reasoning to more than two events.

#### D.1.2  STEP 2: GENERAL CASE FOR THREE EVENTS

Next, we consider the union of three events $A_1, A_2, A_3$. Again, by the inclusion-exclusion principle, we can write:

$$\begin{aligned}P(A_1 \cup A_2 \cup A_3) =&P(A_1) + P(A_2) + P(A_3) - P(A_1 \cap A_2) - P(A_1 \cap A_3) \\ &- P(A_2 \cap A_3) + P(A_1 \cap A_2 \cap A_3)\end{aligned} \tag{39}$$

As before, all intersection terms are non-negative, i.e., $P(A_1 \cap A_2) \geq 0$, $P(A_1 \cap A_3) \geq 0$, $P(A_2 \cap A_3) \geq 0$, and $P(A_1 \cap A_2 \cap A_3) \geq 0$. Thus, we have the following inequality:

$$P(A_1 \cup A_2 \cup A_3) \leq P(A_1) + P(A_2) + P(A_3) \tag{40}$$

This confirms the Union Bound for three events.

#### D.1.3  STEP 3: GENERAL CASE FOR $n$ EVENTS

We now extend this reasoning to $n$ events. Let $A_1, A_2, \ldots, A_n$ be events. We aim to show:

$$P\left(\bigcup_{i=1}^{n} A_i\right) \leq \sum_{i=1}^{n} P(A_i) \tag{41}$$

Using the property of monotonicity, we know that:

$$P\left(\bigcup_{i=1}^{n} A_i\right) = P(A_1 \cup \left(\bigcup_{i=2}^{n} A_i\right)) \tag{42}$$

Applying the inclusion-exclusion principle recursively, we can extend the argument to any finite number of events:

$$P(A_1 \cup (A_2 \cup \cdots \cup A_n)) = P(A_1) + P\left(\bigcup_{i=2}^{n} A_i\right) - P\left(A_1 \cap \left(\bigcup_{i=2}^{n} A_i\right)\right) \tag{43}$$

Since $P\left(A_1 \cap \left(\bigcup_{i=2}^{n} A_i\right)\right) \geq 0$, we have:

$$P(A_1 \cup (A_2 \cup \cdots \cup A_n)) \leq P(A_1) + P\left(\bigcup_{i=2}^{n} A_i\right) \tag{44}$$

Now, by applying this same logic to the remaining $n-1$ events, we continue to decompose the union step by step:

$$P\left(\bigcup_{i=2}^{n} A_i\right) \leq P(A_2) + P\left(\bigcup_{i=3}^{n} A_i\right) \tag{45}$$

Repeating this process for all events, we get:

$$P\left(\bigcup_{i=1}^{n} A_i\right) \leq P(A_1) + P(A_2) + \cdots + P(A_n) \tag{46}$$

Thus, the Union Bound holds for any finite number of events.

### D.1.4 STEP 4: FORMAL PROOF BY INDUCTION

To formalize the argument, we will use mathematical induction.

**Base Case:** For $n = 2$, as shown in Step 1, the Union Bound holds:

$$P(A_1 \cup A_2) \leq P(A_1) + P(A_2) \tag{47}$$

**Inductive Step:** Assume that the Union Bound holds for $n$ events. That is, assume:

$$P\left(\bigcup_{i=1}^{n} A_i\right) \leq \sum_{i=1}^{n} P(A_i) \tag{48}$$

We need to prove that the Union Bound holds for $n + 1$ events, i.e., for $A_1, A_2, \ldots, A_{n+1}$, we need to show:

$$P\left(\bigcup_{i=1}^{n+1} A_i\right) \leq \sum_{i=1}^{n+1} P(A_i) \tag{49}$$

We can write:

$$P\left(\bigcup_{i=1}^{n+1} A_i\right) = P\left(\left(\bigcup_{i=1}^{n} A_i\right) \cup A_{n+1}\right) \tag{50}$$

By the inclusion-exclusion principle, we know:

$$P\left(\left(\bigcup_{i=1}^{n} A_i\right) \cup A_{n+1}\right) = P\left(\bigcup_{i=1}^{n} A_i\right) + P(A_{n+1}) - P\left(\left(\bigcup_{i=1}^{n} A_i\right) \cap A_{n+1}\right) \tag{51}$$

Since $P\left(\left(\bigcup_{i=1}^{n} A_i\right) \cap A_{n+1}\right) \geq 0$, we get:

$$P\left(\left(\bigcup_{i=1}^{n} A_i\right) \cup A_{n+1}\right) \leq P\left(\bigcup_{i=1}^{n} A_i\right) + P(A_{n+1}) \tag{52}$$

By the inductive hypothesis:

$$P\left(\bigcup_{i=1}^{n} A_i\right) \leq \sum_{i=1}^{n} P(A_i) \tag{53}$$

Thus, we have:

$$P\left(\left(\bigcup_{i=1}^{n} A_i\right) \cup A_{n+1}\right) \leq \sum_{i=1}^{n} P(A_i) + P(A_{n+1}) = \sum_{i=1}^{n+1} P(A_i) \tag{54}$$

This completes the inductive step.

### D.2 CONCLUSION

By induction, we have proven that the Union Bound holds for any finite number of events $A_1, A_2, \ldots, A_n$. This result shows that the probability of the union of events

## E SUBSTITUTE MODEL ARCHITECTURE

Our substitute model comprises 12 transformer blocks, each with 768 hidden units and 12 self-attention heads. Each transformer block consists of the following substructures:

- **Self-Attention Layer:** The hidden size of the self-attention layer is 768.
- **Position-wise Feed-Forward Network:** The network first projects the output of the attention layer to a 3072-dimensional space using a fully connected layer, followed by a ReLU activation for non-linearity, and finally projects the 3072-dimensional space back to a 768-dimensional space via another fully connected layer.
- **Layer Normalization and Residual Connection:**
  - **Layer Normalization:** Applied to the output of each sub-layer to stabilize training.
  - **Residual Connection:** Adds the normalized output to the input of the sub-layer.

## F DETAILS OF DATA

Table 7: The statistics of datasets.

| Dataset | Train | Test | classes | Labels name |
|---------|-------|------|---------|-------------|
| SST5 | 8544 | 2210 | 5 | Very positive, Positive, Neutral, Negative, Very negative |
| Emotion | 16000 | 2000 | 6 | Sadness, Joy, Love, Anger, Fear, Surprise |

## G URL

Table 8: Details of the methods in the Baselines.

| Model | Url |
|-------|-----|
| Distilbert | https://huggingface.co/joeddav/distilbert-base-uncased-go-emotions-student |
| BERT | https://huggingface.co/bhadresh-savani/bert-base-go-emotion |
| Roberta | https://huggingface.co/bsingh/roberta_goEmotion |
| A | https://huggingface.co/SamLowe/roberta-base-go_emotions |
| B | https://huggingface.co/Prasadrao/xlm-roberta-large-go-emotions-v3 |
| C | https://huggingface.co/SchuylerH/bert-multilingual-go-emtions |
| D | https://huggingface.co/bergum/xtremedistil-l6-h384-go-emotion |

## H DETAILS OF BASELINES

Table 9: The details of the methods employed in the baseline comparisons. The Perturbed Level indicates the target of the attack methods, where "word" denote the specific words targeted for perturbation, and "char" refer to the characters within a word that are altered by the attack method.

(a) Information on the classification attack method used as the baseline.

| Methods | Perturbed Level | Gradient | Soft-labels | Hard-labels | Knowledge |
|---|---|---|---|---|---|
| Bae | Word | ✗ | ✔ | ✔ | black-box |
| Bert-Attack | Word | ✗ | ✔ | ✔ | black-box |
| DWB | Char | ✗ | ✔ | ✔ | black-box |
| FD | Char | ✔ | ✔ | ✔ | white-box |
| Hotflip | Char | ✔ | ✔ | ✔ | white-box |
| SememePSO | Word | ✗ | ✔ | ✔ | black-box |
| TextBugger | Char+Word | ✔ | ✔ | ✔ | white-box |
| TextFooler | Word | ✗ | ✔ | ✔ | black-box |
| CEMA | Char+Word | ✗ | ✗ | ✔ | black-box |

(b) Information on the translation attack method used as the baseline.

| Methods | Perturbed Level | Gradient | Soft-labels | Hard-labels | Knowledge |
|---|---|---|---|---|---|
| Hotflip(Trans) | Char | ✔ | ✗ | ✗ | white-box |
| kNN | Word | ✔ | ✗ | ✗ | white-box |
| Morphin | Word | ✗ | ✗ | ✔ | black-box |
| RA | Word | ✔ | ✗ | ✗ | white-box |
| Seq2Sick | Word | ✔ | ✗ | ✔ | white-box |
| TransFool | Word | ✗ | ✗ | ✔ | black-box |
| CEMA | Char+Word | ✗ | ✗ | ✔ | black-box |