# OpenReview forum: "Few-shot Text Adversarial  Attack for Black-box Multi-task  Learning"
_ICLR.cc/2025/Conference — Submitted to ICLR 2025_

### Official Review · Reviewer_bJnG · 2024-10-27

**Soundness:** 3
**Presentation:** 3
**Contribution:** 3
**Rating:** 5
**Confidence:** 3

**Summary:**

A new adversarial attack is proposed for text based multi-task models. The attack is called, “CEMA” or Cluster and Ensemble Mutil-task Text Adversarial Attack. Substitute models are trained in this attack and then various adversarial examples are generated. After selecting the example that fools a majority of the models, this example is used as the final attack input. The two task that are attacked are text classification and translation. Empirically the methods are test on two datasets, SST5 and Emotion.

**Strengths:**

The paper is well written and clearly organized. The adversarial threat model is well defined and I appreciate the authors try to use a black-box approach (as opposed to many adversarial attacks that assume complete white-box knowledge of the setup). The empirical results demonstrate the effectiveness of the new method on two datasets.

**Weaknesses:**

There are three main issues that I see with the paper:

Issue 1: Lack of proper citations. There are a number of works that this paper implicitly builds off of but are not explicitly cited. Due to this several experiments are missing that I think would better illustrate and help the readers understand the effectiveness of the proposed approach. First of all the discussion of transferability only mentions the original Papernot work, but is missing other type of attacks that increase the amount of data available to the attacker (https://arxiv.org/abs/2006.10876) or generate synthetic data (https://arxiv.org/abs/2003.12703). I would say at a minimum these two additions to the original 2016 Papernot transfer attack should be cited in relation to improving the CEMA and discussed. Likewise, when discussing transferability in general there are several related and follow up works that the authors do not include any mention of:

https://arxiv.org/abs/1704.03453

https://arxiv.org/abs/2104.02610


Issue 2: Because the authors seems to be unaware of some of the more recent develops in adversarial attacks/defenses, it would be great if two additional experiments or discussions could be had. First: what happens if the query number or amount of training data available to the attacker is increased like in (https://arxiv.org/abs/2006.10876)? I think that is important to understand what effect changing the strength of the adversarial threat model has on the attack success rate. For example if say 1000 or even 10,000 samples of training data are made available to the adversary, how much more effective is the transfer attack? Likewise, if we drop the number down to 50 or 10 samples, what happens? I am okay accepting the paper even IF the attack fails when less samples are used. I just think it is a necessary step in the attack analysis that is currently missing.

Second: Many transfer attacks may fail (citation: https://arxiv.org/abs/2104.02610) when the models are shuffled in a randomized manner. Such work in the image domain that has shown an increased robustness over SOTA single model defenses can be seen here (https://arxiv.org/abs/2211.14669). Because the authors didn’t test on any defenses, I would be curious if they could train additional models and test a simple defense where they try to shuffle between two models during querying and attack time? Or if the authors feel this is not relevant, could they include a discussion of this in their paper and cite the appropriate papers as mentioned in this paragraph to explain why?

Issue 3 (minor): Is there any reason that the authors only choose to use 3 different victim models when doing the experiments? As a reviewer I cannot mandate that additional experiments are done, but I would be curious as to the justifications for three models used in the paper?

**Questions:**

Please address the three issues that I have brought up in the weakness section of the paper. If you can provide better discussions and citations to the relevant adversarial literature and address my comments, I would be inclined to increase my score accordingly.

UPDATE: Score has been updated according to the rebuttal from the authors.

---

> ### Author Response · Authors · 2024-11-28
>
> We thank the reviewer for the valuable comments on our paper. Below is a detailed response to the weaknesses and questions the reviewer mentioned.
>
> **Weaknesses 1.1**
> We include a discussion of these works in Section 2.1 of the paper. Furthermore, we provide an overview of related work on transfer attacks in Section A.4 of the appendix. Please review the updated version of the paper.
>
> **Weaknesses 1.2**
> In Section K of the Appendix, we investigate the impact of varying query numbers and the amount of available training data. Specifically, we set the query numbers and available training data to 10, 50, 100, 1000, and 2000, respectively. Notably, 100 queries correspond to the value used by CEMA in the original paper. Given that a query number of 10,000 is excessively large, we limit the maximum query number to 2000. The results of these experiments are presented in Table 12 of Section J. Our findings indicate that the attack effectiveness increases with the number of queries. Even with only 10 queries, CEMA achieves an attack success rate exceeding 30%.
>
> Additionally, in Section 5.7, we discuss the attack scenario involving zero-shot training data. In this scenario, the attacker cannot access the training data but is aware of its relevant attributes. For example, both the SST5 and Emotion datasets are related to sentiment analysis, but their label spaces and distributions differ significantly. To test this, we used $100$ unlabeled texts from the Emotion validation set as auxiliary data for the SST5 attack, and vice versa.
> The experimental results, presented in Table \ref{data-data}, show that even with limited auxiliary data and significant differences between the auxiliary data and victim texts, CEMA achieves an attack success rate of $66.40\%$ and a BLEU score of $0.27$.
> The findings indicate that an attacker requires only partial knowledge of the training dataset and gathers the relevant data from the Internet. Utilizing CEMA, they can execute a substantial attack on the multi-task system.
>
> **Weaknesses 2**
> We implement a defense method in Section Q of the appendix by utilizing shuffle between two models during both querying and attack phases. CEMA employs three attack methods: DWB, FD, and Textbugger. However, due to the complexity of the code for FD and Textbugger attacks and the requirement to shuffle between two substitute models, it is challenging to complete the necessary code for these attacks within the rebuttal timeframe. As a result, we select DWB and TextFooler as the attack methods for CEMA and enable both methods to shuffle between two models during the querying and attack phases.
>
> The experimental results, presented in Table 19 of Section Q in the appendix, show that while Random Shuffle reduces CEMA's attack effectiveness, CEMA still achieves reasonable performance under this defense.
>
> Additionally, we present two other defense methods—Preceding Language Modifier and Adversarial Training—in Section P.
>
> **Weaknesses 3**
> We introduce three victim models in Sections I, J, and P.2 of the appendix. The victim models in Section I comprise six tasks, including four text classification tasks and two text translation tasks. Section J presents three tasks: text translation, summarization, and text-to-image generation. Section P.2 features two text classification tasks. The attack results of CEMA on these models are provided in Sections I, J, and P.2, demonstrating that CEMA achieves significant attack performance across all three victim models.

---

> ### Author Response · Authors · 2024-11-30
> **Willing to further clarify your remaining concerns**
>
> We appreciate your invaluable time and insightful comments. We have provided more details to your questions and concerns and added experiments. Can you kindly check them and let us know if they address your concerns? If you have further comments, we are happy to have a discussion.
>
> Thank you very much!

---

> > ### Comment · Reviewer_bJnG · 2024-12-01
> >
> > Based on the authors comments I will update my score accordingly, however I will not be doing any further advocating for the paper.

---

### Official Review · Reviewer_3YdR · 2024-10-28

**Soundness:** 4
**Presentation:** 3
**Contribution:** 3
**Rating:** 6
**Confidence:** 2

**Summary:**

This paper presents CEMA, a new text adversarial attack against multi-task learning in the black-box setting. The basic idea of this method is to adopt a cluster-oriented substitute model training to convert the problem into a text classification attack. Representation learning is used for the above clustering. Multiple substitute models are generated to produce the high-quality adversarial examples that are effective against different tasks concurrently. Comprehensive evaluations are performed to demonstrate the effectiveness of this solution.

**Strengths:**

1. This paper is well-written and easy to follow.
2. The proposed method is novel and solid.
3. Experiments are extensive, with baseline comparisons over different methods.

**Weaknesses:**

Thanks for submitting this paper to ICLR. Generally I enjoy reading this paper. It is very clear. I have the following doubts regarding the proposed method.

1. While the attack is efficient in terms of model query, I am wondering what is the computation overhead at local? The attacker requires to train multiple substitute models for candidate selection. However, I could not find the number of substitute models used in this paper. Clearly a larger number of substitute models lead to higher overhead. What is the cost of performing the presentation learning, clustering, training substitute models, and generating adversarial examples?

2. The experiments only consider two types of tasks with three models (two for classification and one for translation). What is the scalability and effectiveness of the method for more types and numbers of tasks?

3. This paper does not consider the evaluation of SOTA defenses against the proposed method. It is recommended to present the attack results under different defense methods, like adversarial training and others.

**Questions:**

1. What is the computation overhead of your attack?

2. What is the scalability of your attack in terms of number of tasks?

3. What is the performance of your attack against SOTA defenses?

---

> ### Author Response · Authors · 2024-11-28
>
> We thank the reviewer for the valuable comments on our paper. Below is a detailed response to the weaknesses and questions the reviewer mentioned.
>
> ### **Weaknesses 1 and Questions 1**
> We train five substitute models on a server equipped with a 24GB NVIDIA 3090 GPU. Each model is trained over two epochs using a dataset containing 100 samples. The training time for a single model is approximately 4 minutes, and the size of each trained model is 418 MB. We added a section on the computation overhead of substitute model training in Section E of the Appendix.
>
> ### **Weaknesses 2 and Questions 2**
> We expand the number of tasks to six downstream tasks, comprising four classification tasks and two translation tasks. The corresponding experimental results are presented in Tables 10 and 11, located in Sections H and I of the appendix. We observe that CEMA achieves an ASR of over 60% on both the SST5 and Emotion datasets, with all BLEU scores below 0.3. These results suggest that the CEMA method can be effectively extended to multi-task learning systems with a broader range of tasks.
>
> Furthermore, we expand the task variety to include three additional types: translation, summarization, and text-to-image generation. CEMA demonstrates consistently significant attack performance, highlighting the adaptability of our method to a broader range and larger number of tasks. Detailed results for the six downstream tasks are presented in Section I of the Appendix, while the outcomes for the additional tasks are included in Section J.
>
> ### **Weaknesses 3 and Questions 3**
> We initiate an extensive exploration of defensive strategies to counter CEMA. In practical systems, we thoroughly investigate various defense mechanisms, including train-free adjustments (Preceding Language Modifier) and adversarial training. The victim models used in our study are after-trained models sourced from the Huggingface website, Ali Translator, and Baidu Translator. Since the training details of these pre-trained models are not publicly available, re-training them using adversarial training is infeasible. Consequently, we adopt training-free defense methods. Specifically, we implement the same approach proposed by [1] and apply prompt learning techniques to large language models (LLMs) to mitigate adversarial text inputs. For this, we provide LLMs with the following prompt:
>
> > "Please revise the following text for grammatical errors, improve the spelling, grammar, clarity, concision, and overall readability."
>
> The results are presented in Table 17. The attack effectiveness of CEMA decreases significantly after employing this defense method. However, the reduced attack performance remains noteworthy, even with an ASR still reaching 40%.
>
> In addition, we train four classification models as victim models and conduct adversarial training to evaluate the impact of adversarial training on CEMA's attack effectiveness. All four models are based on the BERT architecture and are labeled Bert1, Bert2, Bert3, and Bert4. Specifically, Bert1 and Bert3 are trained on the SST5 dataset, while Bert2 and Bert4 are trained on the Emotion dataset. The results of adversarial training are provided in Table 18, Section O of the Appendix. Our findings indicate that although adversarial training reduces CEMA's attack effectiveness, it still demonstrates considerable performance.

---

> > ### Author Response · Authors · 2024-11-30
> > **Willing to further clarify your remaining concerns**
> >
> > We appreciate your invaluable time and insightful comments. We have provided more details to your questions and concerns and added experiments. Can you kindly check them and let us know if they address your concerns? If you have further comments, we are happy to have a discussion.
> >
> > Thank you very much!

---

### Official Review · Reviewer_ovMt · 2024-11-02

**Soundness:** 3
**Presentation:** 3
**Contribution:** 2
**Rating:** 5
**Confidence:** 4

**Summary:**

In this work, the authors propose a transfer-based black-box text attack for multi-task learning called Cluster and Ensemble Multi-task Text Adversarial Attack (CEMA). In particular, CEMA employs a cluster-oriented substitute model training to obtain the substitute mode. Then CEMA generates the adversarial examples using various adversarial attacks and chooses the adversarial example that attacks the most substitute models as the final attack output. Experiments validate the effectiveness of the proposed method.

**Strengths:**

1. The cluster-oriented substitute model training is interesting.

2. The proposed CEMA exhibits good transferability.

3. The authors have derived the theoretical lower bound for CEMA's success rate.

**Weaknesses:**

1. The paper is not well written and hard to follow. It is not clear how to generate adversarial examples using the surrogate model.

2. What is the main contribution of this paper? Is it the substitute model training or the attack method? I do not get the attack method. But if it was the substitute model training, why do not the authors compare CEMA with the adversarial examples generated on various substitute models?

3. Most of the baselines are black-box attack, which can directly attack online APIs. Since the authors evaluate the transferability, I think the authors should consider more efficient attacks using gradient [1,2] as in the image domain.

4. It is weird that the average number of queries is smaller than 1.

[1] Wang, X., Yang, Y., Deng, Y., & He, K. Adversarial training with fast gradient projection method against synonym substitution based text attacks. AAAI 2021.

[2] Xinshuai Dong, Anh Tuan Luu, Rongrong Ji, Hong Liu. Towards Robustness Against Natural Language Word Substitutions. ICLR 2021.

**Questions:**

See weakness

---

> ### Author Response · Authors · 2024-11-28
>
> We thank the reviewer for the valuable comments on our paper. Below is a detailed response to the weaknesses and questions the reviewer mentioned.
>
> ### **Weaknesses 1**
> The substitute model functions as a white-box model, which the attacker can access indefinitely, enabling them to obtain any information necessary for the attack. This unrestricted access allows the attacker to gather the required details and generate adversarial examples capable of successfully deceiving the substitute model. Given that adversarial examples exhibit transferability, those that successfully attack the substitute model are then used to target the victim model.
>
> In this context, **transferability** refers to the ability of adversarial examples generated for one model to produce similar attack effects on different models or systems. Specifically, it means that an adversarial example created for one model can cause an unseen target model to make incorrect predictions or classifications. Even if the target model differs in architecture, training data, or optimization methods from the source model, the adversarial examples can still "transfer" and have an effect on the target model.
>
> The process of crafting adversarial examples using the substitute model is known as a **transfer attack**. We provide a detailed explanation of this process in Appendix L. Additionally, we summarize the concepts of transfer attacks and transferability in Section 2.1.
>
> ### **Weaknesses 2.1 Main Contribution**
> We broaden the scope of text multi-task adversarial attacks through the introduction of **CEMA**, which extends the application of such attacks in several key ways.
>
> 1. **Query Count:** CEMA achieves state-of-the-art (SOTA) attack performance with just 100 queries. Notably, even when the number of queries is reduced to 10, the attack success rate (ASR) remains above 30% (detailed experimental results are provided in Section K of the appendix). To the best of our knowledge, this is the first instance in the adversarial attack community where adversarial examples are successfully generated with such a low query count.
>
> 2. **Black-box Setting:** CEMA operates in a black-box setting, where the attacker only has access to the output texts of the multi-task model. In contrast, most previous methods for text multi-task adversarial attacks operate in white-box settings. Black-box settings, however, are more realistic and practical for real-world scenarios.
>
> 3. **Multi-task Learning Structures:** CEMA explores multi-task learning structures involving multiple distinct tasks. In our original paper, we evaluate CEMA’s performance on text classification and translation tasks. Additionally, Section J of the appendix presents further experiments on summarization and text-to-image generation tasks. Previous research on text multi-task adversarial attacks primarily focuses on attacking multi-task models that handle multiple text classification tasks, whereas our work expands the scope to a wider range of tasks.
>
> 4. **Multi-model Multi-task Learning Systems:** We introduce an attack scenario for **Multi-model Multi-task Learning systems**, a setting that has not been explored in previous adversarial attack research. Past studies primarily focus on parameter-sharing multi-task learning systems, leaving Multi-model Multi-task Learning systems largely unaddressed in the context of adversarial attacks.
>
> Overall, our work proposes four more practical attack scenarios and aims to bridge the gap in existing research on text multi-task adversarial attacks, thereby broadening the scope and application of adversarial attacks in multi-task learning systems.
>
> To address these challenges, we propose **CEMA**, a plug-and-play framework that simplifies the previously mentioned complex attack scenarios into straightforward text classification attacks. Moreover, CEMA adapts to any text classification adversarial method. We also present our attack scenarios as four key research questions in lines 55-60 of the paper’s introduction, with our contributions further summarized in lines 88-101 of the introduction.
>
> ### **Weaknesses 2.2: Compare CEMA with the Adversarial Examples Generated on Various Substitute Models**
> We compare our method with approaches that generate adversarial examples using substitute models, and the results are presented in Table 16 of Section O in the appendix. CEMA outperforms adversarial examples generated on various substitute models. Specifically, CEMA achieves an attack success rate (ASR) that is 30% higher than the second-best ASR, and its BLEU score is 0.15 higher than the second-best BLEU score.

---

> ### Author Response · Authors · 2024-11-28
>
> ### **Weaknesses 3: More Baselines**
> We compare our method with approaches that generate adversarial examples using substitute models, with the results presented in Table 16 of Section O in the appendix. CEMA outperforms adversarial examples generated by various substitute models. Specifically, CEMA achieves an attack success rate (ASR) that is 30% higher than the second-best ASR and a BLEU score that is 0.15 higher than the second-best BLEU score.
>
> ### **Weaknesses 4**
> For each dataset, 100 queries are performed per task, with the SST5 dataset containing 2,210 texts and the Emotion dataset containing 2,000 texts, resulting in an average of 0.045 and 0.05 queries per task, respectively.
>
> To provide further clarification, we have computed the total number of queries for each attack method, as shown in Table 13 and Table 14 of Section M in the appendix. If you prefer to use the total number of queries, we can replace the data in these tables accordingly.

---

> > ### Author Response · Authors · 2024-11-30
> > **Willing to further clarify your remaining concerns**
> >
> > We appreciate your invaluable time and insightful comments. We have provided more details to your questions and concerns and added experiments. Can you kindly check them and let us know if they address your concerns? If you have further comments, we are happy to have a discussion.
> >
> > Thank you very much!

---

### Official Review · Reviewer_RCPA · 2024-11-04

**Soundness:** 1
**Presentation:** 2
**Contribution:** 1
**Rating:** 1
**Confidence:** 4

**Summary:**

This paper is about a new text-based adversarial attack that works simultaneously against multiple NLP models.

**Strengths:**

- The table is easy to read.

**Weaknesses:**

- No real threat model. The authors did not properly specify adversarial capabilities (e.g., number of tokens allowed to flip), so I am unable to interpret what the results are supposed to mean. Note that, without constraints on perturbation of the input, it is trivial to fool sentiment models by providing a movie review with the opposite sentiment, or fool translation models by inputting gibberish. **Please explicitly define the constraints on input perturbations, such as the maximum number of tokens that can be modified or any other relevant limitations on the adversarial capabilities.** This would allow for a more meaningful interpretation of the results.
- The author claims to "demonstrate the effectiveness of CEMA through rigorous mathematical derivations", but the math makes no sense.
  - For example, the authors attempted to justify why generating multiple adversarial examples is useful. They made an argument that each attack query has at least probability $p_\min$ of being successful, so the likelihood of some query being successful is 1 with sufficiently many queries. This is clearly false. Even if $p_\min$ is non-zero, different attack queries would not be independent random variables and the derivation is just wrong.
  - I'm not sure why there is a derivation of the "maximum entropy distribution" (Appendix C) and "union bound" (Appendix D) in the paper. They seem completely irrelevant.
  - Appendix B on transferrability of adversarial examples is barely related to the proposed method itself.
- What do you mean by attacking multi-model multi-task learning? How is this different from transferrable adversarial examples? I am concerned that you are redefining a well-studied problem and not properly acknowledging prior literature.

**Questions:**

- What's the unit of queries in Table 1? What does it mean to use 0.05 queries?

---

> ### Author Response · Authors · 2024-11-28
> **We thank the reviewer for the valuable comments on our paper. Here is a detailed response to the weaknesses and questions the reviewer mentioned.**
>
> ### Weaknesses 1
>
> We define the attack constraint as the semantic similarity between the adversarial and original texts. This constraint is motivated by two main considerations. First, in text classification tasks, sentence-level adversarial attacks generate adversarial examples by modifying sentence structures and text styles, often resulting in substantial word changes. As a result, these methods typically use the semantic similarity between the adversarial and original texts as a constraint. Second, in text translation adversarial attacks, the semantic similarity between the adversarial and original texts is also commonly used as the constraint. The similarity of adversarial examples on the SST5 and Emotion datasets is presented in Section L of the appendix, with values of 0.934 and 0.931, respectively. Additionally, Section R of the appendix provides the definitions of adversarial examples for text classification and text translation. Additionally, we assume that the attack methods are effective, and therefore, in the paper, we assume that the success probability of each attack method is greater than 0.
>
> ### Weaknesses 2.1
>
> Since the exact success probability for each attack method is unavailable, we estimate the attack success probability using the frequency of successful attacks for each method. In the table, we report the frequency \( P(AB) \) of both methods successfully attacking, as well as the frequencies \( P(A) \) and \( P(B) \), which correspond to the success rates of Method A and Method B, respectively. Our analysis shows that \( P(AB) \) is approximately equal to \( P(A) \times P(B) \), with the average value of \( P(A) \times P(B) - P(AB) \) being only 0.17%. The detailed experimental results are presented in Table 20 (Section S) of the supplementary materials. We remove the sections on the union bound and the transferability of adversarial examples in the appendix.
>
> Event independence is defined as the assumption that the occurrence of event A does not affect the occurrence of event B. We assume that the success of adversarial examples generated by Method A does not influence the success of adversarial examples generated by Method B, implying that A and B are independent.
>
> However, this assumption of independence is based solely on experimental validation, which may lack sufficient rigor. Consequently, Sections 4.4 and T provide further discussion on the case of non-independence. We find that, even in non-independent scenarios, using more methods to generate adversarial examples increases the likelihood of successfully attacking the victim model.
>
> \textbf{Weaknesses 2.2 and Weaknesses 2.3}
> The maximum entropy distribution is primarily because, during the clustering process, we selected the clustering result that closely approximates a uniform distribution based on the maximum entropy principle. We have provided additional details on the clustering process in the original paper. Furthermore, we have simplified the proof in Section B of the appendix.
>
> \textbf{Weaknesses 3.1：Multi-model Multi-task Learning}
> Multi-model Multi-task Learning (MMMTL) is a machine learning approach that combines Multi-task Learning (MTL) and Multi-model Learning. MTL involves training multiple related tasks using a shared model, allowing the model to optimize multiple objectives by sharing representations and parameters, as seen in tasks like sentiment analysis and text classification. In contrast, Multi-model Learning employs multiple models to solve a problem, leveraging different models' strengths, such as neural networks, decision trees, and support vector machines. MMMTL merges these two concepts, using multiple models to tackle several related tasks while sharing information between models and tasks, thus improving performance across tasks.

---

> > ### Author Response · Authors · 2024-11-28
> >
> > **Weaknesses 3.2: How is CEMA different from transferrable adversarial examples**
> >
> > In transferable adversarial examples, attackers can access a substitute model, which is assumed to be similar to the victim model. However, in our work, no substitute model exists, and we need to train a substitute model using only 100 black-box queries. Moreover, the substitute model we train does not aim to closely resemble the target model. Instead, it demonstrates strong discriminative capability. In our experiments, we further perform 10 queries to generate adversarial examples for a multi-task model, with results presented in Table 12. Additionally, we conduct experiments in a data-free setting, where the attacker only knows certain properties of the training dataset (e.g., the dataset is sentiment-related) but does not have access to the actual data. In this scenario, adversarial examples are generated without direct access to the data. The results of the data-free experiments are shown in Table 6. Overall, our work primarily explores how to conduct adversarial attacks on a black-box multi-task model with limited access to the victim model, without relying on the existence of a substitute model during the attack process.
> >
> > **Weaknesses 3.2: I am concerned that you are redefining a well-studied problem and not properly acknowledging prior literature.**
> >
> >
> > The CEMA approach primarily addresses the following research questions:
> >
> > 1. **Can attackers generate adversarial examples using a black-box multi-task learning model?** CEMA is designed for black-box settings, where the attacker only has access to the output texts of a multi-task model. In contrast, previous multi-task adversarial attack methods typically operate in white-box scenarios. Black-box settings, however, are more practical and realistic for real-world applications.
> >
> > 2. **How can adversarial examples be crafted in a multi-model multi-task learning system?** CEMA targets adversarial attacks within multi-model multi-task learning systems. Previous studies on text adversarial attacks have primarily focused on parameter-sharing multi-task learning systems and have not explored the challenges of multi-model multi-task learning scenarios.
> >
> > 3. **How can adversarial examples be created using few-shot queries?** CEMA demonstrates that only 100 queries are sufficient to achieve state-of-the-art (SOTA) attack performance. Furthermore, when the number of queries is reduced to just 10, the attack success rate (ASR) remains above 30%. Detailed experimental results are provided in Section K of the appendix. To the best of our knowledge, this is the first instance in the text adversarial attack community where adversarial examples are generated with such a low number of queries.
> >
> > 4. **How can adversarial examples be generated in multi-task learning systems encompassing a variety of tasks?** CEMA investigates adversarial attacks in multi-task learning systems that span multiple distinct tasks. In the original paper, we evaluate CEMA's attack performance on text classification and translation tasks. Additionally, Section J of the appendix explores CEMA’s performance on summarization and text-to-image generation tasks. Previous research on multi-task text adversarial attacks has primarily focused on attacking models with multiple text classification tasks.
> >
> > Overall, we tackle four practical attack scenarios and aim to fill gaps in previous research on multi-task text adversarial attacks. Our work broadens the scope of adversarial attack applications in multi-task learning systems. Additionally, we explore a train-free attack scenario, where the attacker can generate adversarial examples by knowing only the relevant attributes of the training dataset (e.g., sentiment-related) without direct access to the data itself. The results of these train-free experiments are presented in Table 6.
> >
> > To our knowledge, these research questions have not been well-studied within the text adversarial attack community. Additionally, due to space limitations in the main text, we place the related work in Appendix A. We believe that we provide appropriate references to past work in the related work section.
> >
> > ## **Questions 1**
> > For each dataset, 100 queries are performed per task, with the SST5 dataset containing 2,210 texts and the Emotion dataset containing 2,000 texts, resulting in an average of 0.045 and 0.05 queries per task, respectively.
> >
> > To provide further clarification, we have computed the total number of queries for each attack method, as shown in Table 13 and Table 14 of Section M in the appendix. If you prefer to use the total number of queries, we can replace the data in these tables accordingly.

---

> > > ### Author Response · Authors · 2024-11-30
> > > **Willing to further clarify your remaining concerns**
> > >
> > > We appreciate your invaluable time and insightful comments. We have provided more details to your questions and concerns and added experiments. Can you kindly check them and let us know if they address your concerns? If you have further comments, we are happy to have a discussion.
> > >
> > > Thank you very much!

---

### Author Response · Authors · 2024-11-28
**General Response**

We sincerely thank the reviewers for their time, positive comments, and constructive feedback, and are encouraged by the comments of **well-written** (R3, R4), **interesting** (R2), **extensive experimental evaluation** (R2, R3), and **novel and effective** (R3, R4).

In summary, we discuss four practical adversarial attack scenarios in text multi-task learning: few-query attacks, black-box attacks, multi-model multi-task learning, and attacks that encompass a variety of tasks. To the best of our knowledge, few-query attacks, multi-model multi-task learning, and black-box attacks involving a variety of tasks in multi-task text learning are being discussed for the first time. Additionally, we investigate the data-free scenario, in which the attacker only has access to the attributes of the multi-task model, and examine how to perform multi-task text adversarial attacks under these conditions.

We also introduce CEMA, a few-shot, black-box attack method. CEMA is a plug-and-play framework that converts multi-task attack scenarios into single-task text classification attacks. CEMA demonstrates significant attack effectiveness with just 10 queries and in the data-free scenario. Experimental results show that CEMA's performance improves as more text classification methods are incorporated, and it retains substantial effectiveness even when defense mechanisms are applied. We extend CEMA to a variety of tasks, including text classification, translation, summarization, and text-to-image generation.

Considering the reviewer’s comments, we updated our manuscript (modified parts highlighted in “red”), with the following additions:

1. **[Computational Overhead of Substitute Model Training (E)]**
   In Appendix E, we expand on the computational costs involved in training the substitute models for CEMA.

2. **[Performance Evaluation on Six Downstream Tasks (I)]**
   In Appendix I, we evaluate the performance of CEMA on six different downstream tasks, including sentiment analysis (SST5 and Emotion datasets) and machine translation.

3. **[Text Summarization and Text-to-Image Generation Tasks (J)]**
   In Appendix J, we further explore the applicability of CEMA on more complex tasks, such as text summarization and text-to-image generation. Using the Pokemon dataset, we perform experiments on translation (BLEU: 0.24), summarization (ROUGE reduction of 47%), and text-to-image generation (CLIP score reduction of 56%).

4. **[Few-Shot Learning and Additional Model Queries (K)]**
   Appendix K presents the evaluation of CEMA in a few-shot learning scenario, where the attack's effectiveness increases with the number of queries and available training data. Even with as few as 10 queries, the attack success rate exceeds 30%, demonstrating CEMA’s ability to work effectively under resource-constrained conditions.

5. **[Transferability of Adversarial Attacks (L)]**
   In Appendix L, we analyze the transferability of adversarial attacks generated by CEMA across different models. Our transferability experiments show that CEMA achieves superior results, significantly outperforming traditional baseline methods. The transferred adversarial examples effectively compromise multiple target models, showcasing the generalization capability of the attack.

6. **[Impact of Query Count on Attack Performance (M)]**
   In Appendix M, we investigate the relationship between the number of queries and the attack success rate.

7. **[Further Baseline Comparisons on Text Classification Tasks (N)]**
   Appendix N provides a more comprehensive comparison of CEMA against several baseline methods such as TextBugger and HotFlip. The experimental results show that CEMA outperforms these baselines.

8. **[Performance on Transfer Attacks (O)]**
   Appendix O delves deeper into the transferability performance of CEMA across different model combinations.

9. **[Defensive Strategies Against CEMA (P)]**
   In Appendix P, we discuss various defense mechanisms, such as random perturbations, designed to mitigate CEMA’s effectiveness. Although these defenses reduce the attack’s success rate, CEMA still maintains a relatively high performance, with ASR exceeding 50% .

10. **[Adversarial Sample Definition in Text Classification and Translation (R)]**
    In Appendix R, we formally define the properties of adversarial samples used in text classification and machine translation tasks.

11. **[Independence Validation Experiments (S)]**
    Appendix S focuses on validating the independence assumption of the adversarial examples. We show that combining multiple attack methods improves the overall attack success rate, confirming that diverse approaches contribute to more robust adversarial examples.

12. **[Non-Independence in Adversarial Example Generation (T)]**
    In Appendix T, we extend the analysis to scenarios where adversarial examples are not generated independently.

---

### Meta-Review · Area_Chair_MGLp · 2024-12-11

**Metareview:**

- This paper explores *multi-task textual adversarial attacks* and proposes a new attack method, CEMA, to craft multi-task adversarial prompts. The approach involves training substitute models to capture the representation distribution based on cluster labels obtained through limited queries. Multiple attack strategies are combined to enhance task transferability.

- Major concerns raised by reviewers include the realism and meaningfulness of the proposed threat model, the novelty of the CEMA method, and weak experiments. The threat model was revisited during the AC-reviewer process. While multi-task adversarial attacks, in general, are realistic, the focus on attacking only text classification and translation tasks does not sufficiently demonstrate the significance of the approach.  LLMs are inherently multi-task models. This raises questions about the distinction between adversarial attacks on LLMs and the definition of "multi-task" (or "multi-domain") in the context of LLMs. There are also concerns about the adaptability of the approach when incorporating additional task types.

- Overall, the paper cannot be accepted in its current form. I encourage the authors to address the remaining concerns in future submissions.

**Additional Comments On Reviewer Discussion:**

The threat model was revisited during the AC-reviewer process. Reviewers expressed differing opinions regarding the realism of the threat model. The AC believes the threat model is practical and realistic; however, the authors did not effectively prove this point in their experiments.

---

### Decision · Program_Chairs · 2025-01-22

Reject